



**Contributions of primary anthropogenic sources and rapid secondary transformations to organic aerosol pollution in Nanchang, Central China**

Wei Guo[a, b], Zicong Li[a, b], Renguo Zhu[a, b], Zhongkui Zhou[a], Hongwei Xiao[c], Huayun Xiao[c]

a. School of Water Resources and environmental Engineering, East China University of Technology, Nanchang 330013, China

b. Jiangxi Province Key Laboratory of the Causes and Control of Atmospheric Pollution, East China University of Technology, Nanchang 330013, China

c. School of Agriculture and Biology, Shanghai Jiao Tong University, Shanghai 200240, China

Correspondence: Huayun Xiao (xiaohuayun@sjtu.edu.cn)

**Abstract**

Owing to the complex composition of organic aerosols (OAs), it is challenging to elucidate their sources and dynamics, particularly in urban environments in China, where natural and anthropogenic influences converge. We attempted to clarify the relative contributions of primary emissions and secondary formations to urban OAs and confirm the sources and influencing factors of OA pollution. To achieve this, we conducted a comprehensive analysis of major polar organic compounds in fine particulate matter ($PM_{2.5}$) samples collected over a year in Nanchang, Central China. The results indicated that the concentrations of fatty acids, fatty alcohols, and saccharides were relatively high, whereas lignin and resin products, sterols, glycerol, hydroxy acids, and aromatic acids were present at low concentrations. An analysis of molecular characteristics and concentration ratios revealed that they originate from anthropogenic and natural sources. Using the tracer-based method, we observed that the primary organic carbon (POC) and primary organic aerosols (POA) contributed 53% of OC and 21% of $PM_{2.5}$ mass, respectively, compared with a mere 8% and 4% from secondary organic carbon (SOC) and secondary organic aerosols (SOA). Anthropogenic sources were the most dominant determinant, contributing approximately 89% of POC and POA and 60% of SOC and SOA. Seasonal variations indicated that biogenic emissions exerted a stronger influence during spring and summer, whereas anthropogenic emissions were more pronounced in autumn and winter. Short-term winter pollution



episodes were characterized by rapid secondary transformation, promoted by elevated primary emissions and
favorable oxidation conditions, including increased light intensity and nitrogen oxides.
**1 Introduction**
Fine particulate matter (PM), specifically those with an aerodynamic diameter less than 2.5 μm ($PM_{2.5}$), affects
the environment and climate, posing severe threats to human health (Huebert et al., 2003; Kanakidou, 2005;
Riipinen et al., 2012; Shiraiwa et al., 2017; Yazdani et al., 2021). This is evident in China, which has become a
focal point for $PM_{2.5}$ pollution, characterized by recurrent haze episodes that have intensified since 2011 (Cao
et al., 2012; Zhang et al., 2012; Huang et al., 2014). To tackle this pressing challenge, the Chinese government
implemented the "Air Pollution Prevention and Control Action Plan" in 2013, revisions to the "Air Pollution
Prevention and Control Law" in 2014, and the "Three-year Action Plan for Winning the Blue Sky Defense
War" in 2018. Such initiatives have significantly reduced $PM_{2.5}$ levels, with concentrations below 35 μg m$^{-3}$ as
of 2020. However, the challenge persists, particularly during winter, as exemplified by the winter of 2023–
2024, when $PM_{2.5}$ concentrations in many cities exceeded the national air quality standards (Figures 1a and S1).
This underscores an urgent need for effective air quality management strategies (An et al., 2019; Cao and Cui,
2021; Chen et al., 2021; Wu et al., 2021; Zhang et al., 2024). To address urban $PM_{2.5}$ pollution, a
comprehensive investigation of pollutant components, an identification of pollution sources, and an evaluation
of influencing factors are required.
Organic components make up an important part of $PM_{2.5}$, accounting for 30%–50% of its mass, with
primary and secondary sources (Huang et al., 2014; Zhang et al., 2016; Haque et al., 2022). Primary organic
carbon (POC) or primary organic aerosol (POA) is directly emitted from various sources, including biomass
burning, coal combustion, vehicle exhaust, cooking, plant debris, and fungal spores (Guo et al., 2012; Yazdani
et al., 2021). Secondary organic carbon (SOC) or secondary organic aerosol (SOA) forms in the atmosphere
through the photooxidation of biogenic and anthropogenic volatile organic compounds (BVOCs and AVOCs)
(Lewandowski et al., 2008; Stone et al., 2009). Common biogenic precursors include hemiterpenes,
monoterpenes, and sesquiterpenes, whereas typical anthropogenic precursors include toluene and polycyclic
aromatic hydrocarbons (PAHs) (Claeys et al., 2004; Al-Naiema and Stone, 2017; Jaoui et al., 2019). Owing to



the complex composition of OAs, their diverse emission sources, and the impact of meteorological conditions
and photochemical oxidation processes, the identification of OA sources is complicated (Ding et al., 2013;
Zhang et al., 2024). For years, research on the composition and source apportionment of OAs has attracted
considerable attention. However, a unified conclusion regarding OA sources, particularly in China's intricate
urban environments influenced by natural and anthropogenic factors, remains elusive (Wang et al., 2006; Fu et
al., 2008; Guo et al., 2012; Ding et al., 2014; Xu et al., 2022). Various source apportionment methods have
been employed. Such methods include the elemental carbon and water-soluble OC methods (EC-based and
WSOC-based method; Xu et al., 2021), compound tracer method (tracer-based method; Ding et al., 2014; Ren
et al., 2021; Haque et al., 2023), isotope signature method (Tang et al., 2022; Xu et al., 2022 and 2023; Zhang
et al., 2024), and source apportionment models, such as the chemical mass balance (CMB) and positive matrix
factorization models (Xu et al., 2021; Zhang et al., 2023). However, these methods yielded different source
contributions. The relative significance of primary emissions versus secondary formation in urban OAs
continues to be controversial. Further investigation is required to clarify the sources of OA pollution and the
influencing factors.

Thus, this paper presents a comprehensive assessment of OAs in Nanchang, a city in Central China

characterized by moderate $PM_{2.5}$ pollution levels that reflect broader urban atmospheric conditions across
China. Employing the EC- and tracer-based methods, as well as the CMB model, we quantitatively evaluated
the contributions of POC and SOC to OC, as well as the contributions of the corresponding POA and SOA to
the mass of $PM_{2.5}$. The results showed that urban OAs were predominantly influenced by anthropogenic
sources, with primary contributions exceeding secondary contributions. Based on a continuous year-long
observation (November 1, 2020 to October 31, 2021, 365 daily samples), we discovered that the anthropogenic
contributions significantly increased in autumn and winter. However, the biogenic contributions increased in
spring and summer. Short-term winter pollution episodes were promoted by rapid secondary transformation,
primarily due to high primary emissions and favorable oxidation conditions, including increased light intensity
and nitrogen oxides ($NO_x$). This study integrates multiple source apportionment methods and accounts for the
seasonal characteristics of OA pollution, providing a robust framework for understanding urban air quality
dynamics. The findings have implications for governmental strategies for mitigating air pollution and





preventing haze episodes in Nanchang and its environs. The insights obtained serve as a future reference for
investigating the impacts of organic compounds in $PM_{2.5}$ on visibility, public health, and climate change.
**2 Materials and methods**
**2.1 Sampling sites and sample collection**
The study area was Nanchang, Central China, with sampling at the East China University of Technology
(ECUT) located in the northwest region of the city (28.72°N, 115.83°E; Figure 1b). As detailed in our previous
investigations (Guo et al., 2024a and 2024b), the sampling site was in a mixed-use area characterized by
educational, commercial, transportation, and residential activities, devoid of significant local pollution sources.
The sampler was strategically placed on the rooftop of a six-story building, approximately 20 m in height,
ensuring an unobstructed sampling environment.
Sampling was conducted from November 1, 2020, to October 31, 2021, with daily $PM_{2.5}$ sample collection.
An 8-inch × 10-inch quartz fiber filter (Pall Tissuquartz, USA) was used in a high-volume air sampler for
sample collection. The meteorological parameters and gaseous pollutant data for the sampling period were
sourced from publicly available online monitoring platforms (https://weatherandclimate.info and
http://www.aqistudy.cn/). Text S1 and Figure S2 provide a comprehensive overview of the prevalent
meteorological conditions and air quality during the sampling period.
**2.2 Chemical analyses**
The OC and EC concentrations in the $PM_{2.5}$ samples were quantified using the Desert Research Institute Model
2001 Carbon Analyzer, following the thermal/optical reflectance protocol established by the Interagency
Monitoring of Protected Visual Environments (IMPROVE). A 1.0-cm² filter sample was placed in a quartz
boat in the analyzer and subjected to incremental heating at predetermined temperatures. Repeated analyses
were performed, demonstrating an analytical uncertainty of ±10%.
We employed methodologies outlined by Wang et al. (2005, 2006) and Fu et al. (2008, 2010) for the
extraction and derivatization of organic compounds. The filter samples underwent three consecutive ultrasonic
extractions using a dichloromethane–methanol mixture (2:1 v/v), followed by the concentration and drying of
the resulting extracts. Prior to instrumental analyses, N,O-bis-(trimethylsilyl) trifluoroacetamide and pyridine
were introduced to derivatize the polar compounds in the extract, with C13 n-alkanes added as internal



standards for quantitative analyses.
The identification and quantification of organic compounds were achieved via gas chromatography-mass
spectrometry (GC-MS), using a Thermo Scientific TRACE GC coupled to a Thermo Scientific ISQ QD single
quadrupole mass spectrometer. GC separation was performed using a DB-5MS fused silica capillary column.
Text S2 and Table S1 present the parameters for the GC-MS analysis, including temperature elevation
procedures, qualitative and quantitative methods for the compounds, and quality assurance and control
protocols. Polar and nonpolar organic compounds in the extracts were simultaneously analyzed; however, this
study focused on the results regarding the polar compounds; the findings related to nonpolar compounds are
provided in our earlier publications (Guo et al., 2024a and 2024b).
**2.3 Source apportionment methods**
To assess the contributions of various primary and secondary sources to OC and $PM_{2.5}$, we employed the
tracer-based approach, a well-established method for source apportionment. This study focused on two types
of organic tracers that serve as indicators for POC and SOC or POA and SOA. By integrating the reported
ratios of specific tracers to OC and aerosol mass concentrations from various emission sources with the actual
tracer concentrations measured in the present samples, we calculated the contributions of these sources to OC
and $PM_{2.5}$. The calculation equation was as follows (Kleindienst et al., 2007 and 2012):
$$[POC]=\frac{\sum_i[tr_i]}{f_{POC}},$$      (1)

$$[POA]=\frac{\sum_i[tr_i]}{f_{POA}},$$      (2)

$$[SOC]=\frac{\sum_i[tr_i]}{f_{SOC}},$$      (3)

$$[SOA]=\frac{\sum_i[tr_i]}{f_{SOA}},$$      (4)
where $\sum_i[tr_i]$ is the total concentration of the selected tracers in the sample, denoting representative
compounds from specific emission categories; $f_{POC}$ and $f_{POA}$ are the mass fractions of the tracers in OC and





PM$_{2.5}$ from primary emissions, respectively. Similarly, f$_{SOC}$ and f$_{SOA}$ are the mass fractions of the tracers in OC
and PM$_{2.5}$ from secondary emissions, respectively. The calculated [POC], [POA], [SOC], and [SOA] are the
contributions of different primary and secondary sources to OC and PM$_{2.5}$. Here, we identified mass fractions
for representative tracers from several significant primary and secondary emission sources. The primary
sources included biomass burning, coal combustion, vehicle exhaust, cooking activities, plant debris, and
fungal spores. The secondary sources were categorized into biogenic (isoprene, monoterpenes, and
sesquiterpenes) and anthropogenic (toluene and naphthalene) sources. Table S2 presents information on the
representative tracers, their corresponding f$_{OC}$ and f$_{OA}$, and relevant references.
We employed the CMB model (version 8.2) provided by the US Environmental Protection Agency to verify
the results calculated using the tracer-based method. This model assumes a CMB between the emission sources
and environmental receptors. Thus, the mass of pollutants is not lost during transport from the source to the
receptor, and the measured chemical component concentration at the receptor is the linear superposition of the
contribution of each source class to the concentration (Stone et al., 2009). Furthermore, we employed the EC-
based method to estimate the total POC and SOC concentrations, enabling a comparison with the POC and
SOC concentrations obtained via the tracer-based approach. This method uses EC as a tracer for POC and
approximates the observed minimum ratio of OC to EC (OC/EC)$_{min}$ as an equivalent to the primary emission
OC/EC value (OC/EC)$_{pri}$ (Turpin and Huntzicker, 1995; Castro et al., 1999). The equation used for estimating
SOC is
$$SOC_{EC\text{-}based}=OC - EC\times(OC/EC)_{min}. \tag{5}$$
Here, the minimum OC/EC values observed in spring, summer, autumn, and winter were 2.01, 2.20, 2.15, and
2.31, respectively.
**3 Results and discussion**
**3.1 Carbonaceous components**
The OC and EC concentrations were in the ranges of 1.01–13.66 µg m$^{-3}$ (mean: 5.88 ± 2.57 µg m$^{-3}$) and 0.27–
4.55 µg m$^{-3}$ (mean: 1.75 ± 0.88 µg m$^{-3}$), respectively. Seasonal variations in the OC and EC concentrations
closely mirrored those observed for PM$_{2.5}$, with elevated levels in autumn and winter, and diminished levels in
spring and summer (Figure 2). The OC/EC ratio is a widely employed metric for characterizing emissions from

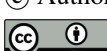



fossil fuels and biomass combustion. OC/EC values below 1.1 typically indicate vehicle exhaust emissions;
values within the range of 2–3 suggest coal combustion emissions, and values above 7 indicate biomass
burning (Saarikoski et al., 2008). Here, the OC/EC range was 2.01–16.95, with an average of 3.86 ± 2.05.
These OC/EC ratios fell within the ranges associated with coal and biomass burning emissions. This suggests
that the carbonaceous material was predominantly derived from combustion. Furthermore, owing to its
stability and resistance to chemical transformations in the atmosphere, EC is frequently used as a tracer for
primary emissions. The OC/EC ratio is a useful tool for assessing the relative contributions of primary and
secondary sources (Castro et al., 1999;Turpin and Huntzicker, 1995). Generally, an OC/EC value above 2
indicates a predominant contribution of secondary sources to OC, whereas values below 2 suggest greater
contributions from primary sources (Kunwar and Kawamura, 2014). Here, the majority of the OC/EC values
exceeded 2, implying that the OC may have been significantly influenced by secondary sources. The OC/EC
values were relatively higher in summer (4.33 ± 2.07) and winter (4.18 ± 2.57) compared with those observed
in spring (3.36 ± 1.98) and autumn (3.59 ± 1.22). This indicates an increased contribution to OC from
secondary sources during the summer and winter.
**3.2 Major polar components**
**3.2.1 Fatty acids**
A range of homologous straight-chain C10:0–C32:0 fatty acids (FAs) and unsaturated C16:1 and C18:1 FAs
were detected in the $PM_{2.5}$ samples (Figure 3a). Their distribution exhibited a strong even carbon number
predominance, as indicated by a Carbon Preference Index (CPI) of 7.59 ± 7.42 (Figure 4a), with peaks at
C16:0 and C18:0. The total FA concentration was in the range of 3.24–657.86 ng m$^{-3}$, with an average
concentration of 196.50 ± 110.92 ng m$^{-3}$. Similar molecular distribution patterns and concentrations have been
documented in urban aerosol studies across China (59–2,090 ng m$^{-3}$; Wang et al., 2006; Fu et al., 2008; Haque
et al., 2019; Fan et al., 2020). This average concentration generally exceeds those observed for coastal and
marine aerosol samples (0.1–160 ng m$^{-3}$; Kawamura et al., 2003; Wang et al., 2007; Fu et al., 2011). High
molecular weight FAs (HFAs, ≥ C20:0) are typically derived from the waxes of terrestrial higher plants or
biomass burning, whereas low molecular weight FAs (LFAs, < C20:0) originate from more diverse sources,
including vascular plants, microorganisms, marine phytoplankton, and kitchen emissions (Fu et al., 2010).





Consequently, the ratio of LFAs to HFAs (LFAs/HFAs) is an indicator of the relative contributions from
various sources. Here, the LFA/HFA range was 0.34–10.67, with an average of 2.45 ± 1.53 (Figure 4b),
suggesting that the FAs may have originated from a mixture of natural and anthropogenic sources. The specific
FAs C16:0 and C18:0 are associated with different sources. C16:0 is primarily derived from biomass burning
or plant emissions, whereas C18:0 predominantly originates from vehicle exhaust, cooking emissions, or road
dust (Wang et al., 2007; Fu et al., 2010). Thus, the ratio of C18:0 to C16:0 (C18:0/C16:0) is frequently used to
assess the FA source. A C18:0/C16:0 ratio below 0.25 suggests a primary contribution from biomass burning or
plant emissions, whereas ratios between 0.25 and 0.5 indicate a predominant influence from vehicle exhaust.
C18:0/C16:0 ratios within the range of 0.5–1 suggest a significant contribution from cooking emissions or road
dust (Rogge et al., 2006). Here, the C18:0/C16:0 varied from 0.05 to 9.53, with an average of 0.59 ± 0.88
(Figure 4c), indicating that the FAs in Nanchang originated from a mixture of sources.

The average concentrations of unsaturated FAs, specifically C16:1 and C18:1, were 6.94 ± 4.35 and 3.79 ±

2.86 ng m$^{-3}$, respectively. Such unsaturated FAs are primarily derived from biomass burning, plant leaf
emissions, cooking activities, and release from marine organisms (Kawamura and Gagosian, 1987; Rogge et
al., 1993; Nolte et al., 1999). Upon emission into the atmosphere, unsaturated FAs undergo rapid oxidation by
ozone ($O_3$), hydrogen peroxide ($H_2O_2$), or hydroxyl (OH) radicals. Consequently, the ratio of unsaturated FAs
to saturated FAs, represented as (C16:1 + C18:1)/(C16:0 + C18:0), is an indicator of the reactivity and degree
of aging of unsaturated FAs (Rudich et al., 2007; Kawamura and Gagosian, 1987). Here, this ratio was 0.12 ±
0.06 (Figure 4d), indicating that the unsaturated FAs underwent significant photochemical degradation and that
secondary organic compounds may be common in PM$_{2.5}$.

The LFAs/HFAs and C18:0/C16:0 were relatively low during autumn and winter compared with spring and

summer (Figures 4b and 4c). This indicated that in autumn and winter, FAs were significantly influenced by
terrestrial plant or biomass burning, whereas in spring and summer, they were more influenced by vehicular
emissions, marine phytoplankton, and cooking emissions. Conversely, the (C16:1 + C18:1)/(C16:0 + C18:0)
values were relatively higher in autumn and winter than in spring and summer (Figure 4d). This suggested that
unsaturated FAs experienced a greater degree of photochemical degradation during the warm seasons.
**3.2.2 Fatty alcohols**




A series of straight-chain n-alkanols (C14–C32) were detected (Figure 3b), exhibiting a predominance of even
carbon numbers, as indicated by a CPI of 10.28 ± 17.05 (Figure 4e). Low molecular weight alcohols (LMW$_{alc}$,
≤ C20) exhibited peaks at C16 and C18, and high MW alcohols (HMW$_{alc}$, > C20), exhibited peaks at C26,
C28, and C30 (Figure 3b). The total concentration of n-alkanols was in the range of 5.80–572.01 ng m$^{-3}$
(average = 113.99 ± 92.50 ng m$^{-3}$). This concentration fell within the range of reported n-alkanol
concentrations in urban aerosols in China (3.1–1,301.0 ng m$^{-3}$; Wang et al., 2006; Fu et al., 2008; Haque et al.,
2019; Fan et al., 2020) and generally exceeded that of coastal and marine aerosol samples (0.1–19.7 ng m$^{-3}$;
Kawamura et al., 2003; Fu et al., 2011). HMW$_{alc}$ mainly originates from higher plant leaf waxes, loess
deposits, or biomass burning, and LMW$_{alc}$ primarily originates from soil and marine microorganisms
(Simoneit, 2002; Kawamura et al., 2003). Here, the LMW$_{alc}$ concentration (75.12 ± 61.28 ng m$^{-3}$) relatively
exceeded that of HMW$_{alc}$ (39.09 ± 34.07 ng m$^{-3}$), yielding an average LMW/HMW of 2.50 ± 1.73 (Figure 4f),
indicating a mixture of n-alkanols from soil, marine organisms, and biomass burning. The LMW/HMW values
were relatively lower in autumn and winter and higher in spring and summer (Figure 4f). This suggests an
increased contribution of soil and marine organisms to n-alkanols in spring and summer. Contributions from
plant or biomass burning increased in autumn and winter.
**3.2.3 Saccharides**
We investigated three saccharide classes: anhydrosugars, primary sugars, and sugar alcohols. Among the
anhydrosugars, levoglucosan had the highest concentration (102.48 ± 78.63 ng m$^{-3}$). We observed markedly
lower concentrations of mannosan (4.33 ± 4.04 ng m$^{-3}$) and galactosan (2.41 ± 2.67 ng m$^{-3}$) (Figure 3c).
Levoglucosan, a prominent biomarker for biomass burning, has been extensively investigated in urban aerosol
samples, with concentrations in the range of 22–2,706 ng m$^{-3}$ (Wang and Kawamura, 2005; Wang et al., 2006;
Fu et al., 2010; Haque et al., 2019; Fan et al., 2020), indicating the significant influence of biomass burning on
the urban atmosphere. The widespread detection of levoglucosan across diverse environments, including
suburban (10–482 ng m$^{-3}$; Yttri et al., 2007; Fu et al., 2008), marine (0.2–30 ng m$^{-3}$; Simoneit et al., 2004a; Fu
et al., 2011), and polar (0–3 ng m$^{-3}$; Stohl et al., 2007; Fu et al., 2009) regions, suggests its potential for long-
range atmospheric transport. Seasonal variations of anhydrosugars exhibited consistent patterns, with elevated
concentrations in autumn and winter and reduced levels in spring and summer (Figure 3c). This indicated

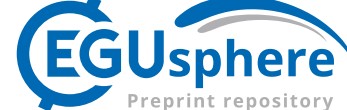



increased biomass burning during the cold months. The sources were characterized using levoglucosan-to-
mannosan (L/M) and mannosan-to-galactosan (M/G) ratios. Research suggests distinct L/M ranges for
different biomass-burning sources: softwood (3–10), hardwood (13–35), and agricultural crop burning (40–56)
(Oros and Simoneit, 2001; Sheesley et al., 2003; Engling et al., 2014). Coal combustion can produce
significant amounts of levoglucosan (Sheesley et al., 2003; Fabbri et al., 2009), typically with higher L/M
values (12–189) because of the lower cellulose content compared with that of plant materials (Rybicki et al.,
2020a and 2020b). Conversely, M/G values are generally higher for softwood and hardwood combustion at
3.6–7.0 and 1.2–2.0, respectively. However, M/G values are relatively lower for agricultural crop burning (0.3–
0.6) (Oros and Simoneit, 2001; Sheesley et al., 2003). Here, most samples exhibited relatively high L/M ratios
(4.63–466.61, average = 40.22 ± 49.82), and relatively low M/G ratios (0.30–9.41, average = 2.34 ± 1.36). The
L/M and M/G ratios were similar to those of previous observations in the urban area of Nanchang (L/M = 7.9–
359.1; average = 59.9; Zhu et al., 2022) and fell within the ranges for crop residue burning and coal
combustion. This suggests that the dehydrosugars in the $PM_{2.5}$ of Nanchang were likely primarily derived from
crop residue burning and coal combustion. In the autumn and winter, the contributions of crop residue burning
and coal combustion were more prominent, as evidenced by the higher L/M ratios and lower M/G ratios
observed compared with those in the spring and summer (Figures 4g and 4h).
Primary sugars and sugar alcohols serve as tracers for primary biological aerosol particles, originating from
biogenic emissions, including the release of microorganisms, plants, and flowers, as well as the resuspension
of surface soils and unpaved road dust containing biological materials (Graham et al., 2003; Simoneit et al.,
2004b;Yttri et al., 2007). Furthermore, biomass burning is a significant source of primary sugars and sugar
alcohols (Fu et al., 2012). We identified nine primary sugars (ribose, fructose, galactose, glucose, sucrose,
lactulose, maltose, turanose, and trehalose; Figure 3c). Glucose ($14.90 \pm 6.62$ ng m$^{-3}$) and sucrose ($14.68 \pm$
$6.48$ ng m$^{-3}$) exhibited the highest concentrations, followed by fructose ($10.82 \pm 4.54$ ng m$^{-3}$), trehalose ($7.92$
$\pm 4.40$ ng m$^{-3}$), and maltose ($5.39 \pm 3.43$ ng m$^{-3}$). The concentrations of galactose ($1.88 \pm 1.24$ ng m$^{-3}$), ribose
($2.87 \pm 2.18$ ng m$^{-3}$), turanose ($0.87 \pm 0.61$ ng m$^{-3}$), and lactulose ($0.83 \pm 0.54$ ng m$^{-3}$) were comparatively
low. We detected four sugar alcohols (mannitol, arabitol, pinitol, and inositol). Mannitol ($9.88 \pm 4.57$ ng m$^{-3}$)
exhibited the highest concentration, followed by pinitol ($6.05 \pm 4.23$ ng m$^{-3}$) and arabitol ($4.96 \pm 3.39$ ng m$^{-3}$).



The concentration of inositol ($2.05 \pm 1.66$ ng m$^{-3}$) was low.
Levoglucosan serves as a specific biomarker for biomass burning, enabling the assessment of biomass-
burning contributions to sugar compounds through the ratio of levoglucosan to OC (Lev/OC) and the
proportion of levoglucosan in the total sugar compounds (Lev%). Typically, a Lev/OC ratio above 0.048 and a
Lev% above 68% indicate a dominance of biomass burning. Conversely, values below these thresholds suggest
that, in addition to biomass burning, other sources, such as biogenic emissions, significantly contribute to the
presence of sugar compounds (Yan et al., 2019 and references therein). Here, the Lev% and Lev/OC were
$53.57\% \pm 17.93\%$ and $0.016 \pm 0.009$ (Figures 4i and 4j), respectively, indicating that the sugar compounds
were influenced by biomass burning and biogenic emissions, particularly the primary sugars and sugar
alcohols. Primary sugars, such as glucose and trehalose, as well as the mannitol in sugar alcohol, positively
correlated with levoglucosan (Figure S3a). This confirmed that biomass burning may be a potential source of
primary sugars and sugar alcohols. However, the correlation was not statistically significant, suggesting that
biological sources remain an important contributor to these compounds in Nanchang. The Lev/OC and Lev%
values were higher in autumn and winter than in spring and summer (Figures 4i and 4j), further indicating that
biomass-burning contributions increased during the cold months.
**3.3 Minor polar components**
**3.3.1 Lignin, resin products, and sterols**
Here, we detected four types of lignin and resin-derived compounds: 4-hydroxybenzoic acid, dehydroabietic
acid, vanillic acid, and syringic acid (Figure 5a). These are generally considered to originate from natural plant
release and burning. The burning of coniferous trees, such as pine, releases high concentrations of
dehydroabietic acid compared with other lignin and resin-derived compounds (Simoneit, 2002). Here, the
concentrations of 4-hydroxybenzoic acid ($5.06 \pm 3.57$ ng m$^{-3}$) and dehydroabietic acid ($2.98 \pm 1.91$ ng m$^{-3}$)
were high, whereas those of vanillic acid ($1.50 \pm 1.08$ ng m$^{-3}$) and syringic acid ($1.53 \pm 1.11$ ng m$^{-3}$) were
relatively low. A significant positive correlation was observed between the total concentration of lignin and
resin-derived compounds and levoglucosan ($r = 0.80$, $p < 0.01$, Figure S3a), suggesting that biomass burning
was a potential source of lignin and resin-derived compounds. The concentration of dehydroabietic acid was
significantly lower than that of 4-hydroxybenzoic acid, similar to the pattern observed for Nanjing aerosols



(Haque et al., 2019). This implies that coniferous tree combustion may not be the primary source of lignin and
resin acids in this region.

We identified several sterols (cholesterol, stigmasterol, and β-sitosterol, average concentrations = 4.63 ±

3.17, 4.89 ± 3.68, and 10.11 ± 8.26 ng m$^{-3}$, respectively; Figure 5a). These sterols originate from distinct
sources. Cholesterol, an animal-derived sterol, is primarily associated with meat cooking in the atmosphere
(Rogge et al., 1991; Nolte et al., 1999; He et al., 2004). Contrarily, stigmasterol and β-sitosterol are plant-
derived sterols, typically originating from plant leaves, cooking, or biomass burning (Nolte et al., 2001; He et
al., 2004; Zhao et al., 2007). Here, stigmasterol and β-sitosterol exhibited a significant positive correlation with
levoglucosan (Figure S3a). This suggests that biomass burning may be a potential source of plant-derived
sterols. The concentration of stigmasterol and β-sitosterol was higher in autumn and winter than in spring and
summer, indicating an increased contribution of biomass burning to sterols in the autumn and winter.

### 305    3.3.2 Glycerol and hydroxy acids

Glycerol and three hydroxy acids, glycolic acid, malic acid, and citric acid, were detected in the PM$_{2.5}$ samples
(Figure 5b). The concentration of glycerol was relatively high (9.99 ± 4.13 ng m$^{-3}$), whereas the three hydroxy
acids exhibited comparable and relatively lower concentrations: glycolic acid (7.16 ± 1.90 ng m$^{-3}$), malic acid
(6.60 ± 1.85 ng m$^{-3}$), and citric acid (5.92 ± 1.99 ng m$^{-3}$). These concentrations aligned with reported ranges
for urban aerosols (2–146 and 1–180 ng m$^{-3}$ for glycerol and hydroxy acids, respectively; Wang et al., 2006;
Fu et al., 2010; Haque et al., 2019; Fan et al., 2020). The glycerol in the aerosol is primarily derived from
fungal metabolism in suspended soil (Simoneit et al., 2004b), whereas glycolic, malic, and citric acids mainly
originate from the secondary photooxidation of organic compounds in the atmosphere (Kawamura and
Ikushima, 1993; Kawamura and Sakaguchi, 1999; Claeys et al., 2004). No significant correlation was observed
between glycerol and the hydroxy acids; however, a notable correlation was observed among the hydroxy acids
(r = 0.49–0.66, p < 0.01, Figure S3b). This confirmed the divergent sources of glycerol and hydroxy acids and
similar origins for the hydroxy acids. In summer, the concentrations of the hydroxy acids exceeded those in
other seasons (Figure 5b). A positive correlation was observed between the polyacids and temperature (r =
0.35–0.51, p < 0.01), suggesting enhanced secondary photochemical oxidation processes in the warm months.

### 320    3.3.3 Aromatic acids



We detected four aromatic acids in the PM$_{2.5}$ samples: benzoic acid, phthalic acid, isophthalic acid, and
terephthalic acid at concentrations of 0.76 ± 0.34, 6.40 ± 4.05, 1.10 ± 0.74, and 10.06 ± 3.07 ng m$^{-3}$,
respectively (Figure 5b). Aromatic acids significantly contribute to the formation of atmospheric particles.
Benzoic acid is recognized as a major pollutant in vehicle exhaust and the secondary photochemical oxidation
product of aromatic hydrocarbons from vehicle emissions (Kawamura and Kaplan, 1987; Rogge et al., 1993;
Kawamura et al., 2000). Phthalic acid typically originates from the secondary transformation of PAHs,
including naphthalene and other PAHs. Terephthalic acid is primarily produced through the hydrolysis of
terephthalate during the combustion of urban plastics (Fu et al., 2010; Haque et al., 2019). Here, phthalic and
terephthalic acids jointly accounted for approximately 90% of the total aromatic acids detected. A significant
positive correlation was observed between phthalic acid and PAHs (r = 83, p < 0.01), as well as between
terephthalic acid and the total phthalates (r = 0.72, p < 0.01). The PAH and phthalate data have been published
by Guo et al. (2024a and 2004b). This suggests that the secondary conversion of PAHs and the hydrolysis of
phthalates (specifically terephthalate) during plastic burning significantly contribute to aromatic acid levels.
The phthalic acid concentrations were higher in autumn and winter, whereas terephthalic acid levels peaked in
spring and summer (Figure 5b). This indicates that the secondary transformation of PAHs was more
pronounced during the cold months, whereas high temperatures in spring and summer promoted the
volatilization and transformation of phthalates from plastic sources.
**3.4 Secondary organic aerosol tracers in PM$_{2.5}$**
**3.4.1 Biogenic secondary organic aerosol tracers**
Biogenic and anthropogenic SOAs play critical roles in influencing the atmospheric radiation balance and
regional air quality (Ding et al., 2014). Biogenic SOA tracers include oxidation products from isoprene,
monoterpenes, sesquiterpenes, and other oxygenated hydrocarbons. Isoprene is the dominant component of
BVOC emissions, accounting for approximately 45% of the total emissions (annual release estimated at 600
Tg C) (Piccot et al., 1992; Guenther et al., 1995; Sharkey et al., 2008). Here, we detected six isoprene
oxidation products (2-methylglyceric acid (2-MGA), three C5-alkene triols, and two 2-methyltetrols (MTLs) at
concentrations of 1.83 ± 1.11, 2.84 ± 1.88, and 4.14 ± 3.27 ng m$^{-3}$, respectively; Figure 6). The concentrations
of the C5-alkene triols and MTLs exceeded that of 2-MGA. Generally, C5-alkene triols primarily originate



from the secondary transformation of VOCs released during biomass burning and higher plant waxes (Fu et al.,
2010 and 2014). Throughout the sampling period, a strong linear correlation was observed between the MTLs
and C5-alkene triols ($R^2 = 0.82$, $P < 0.01$), indicating their similar sources. Nevertheless, certain differences
were observed in the formation processes of C5-alkene triols and MTLs. Both compounds are formed through
the acid-catalyzed ring-opening reaction of isoprene epoxydiols (Surratt et al., 2006 and 2010); however, the
formation of C5-alkene triols is enhanced in acidic aerosol environments (Yee et al., 2020). Here, the ratios of
C5-alkene triols to MTLs (C5/MTLs) were elevated in autumn and winter compared with those observed in
spring and summer (Figure 7a), suggesting that increased aerosol acidity in the cold months may facilitate the
production of C5-alkene triols. C5-alkene triols and MTLs are believed to form via isoprene oxidation under
low-$NO_x$ conditions (Surratt et al., 2010; Lin et al., 2013). Contrarily, high concentrations of $NO_x$ favor the
further oxidation of isoprene to yield 2-MGA (Lin et al., 2013; Nguyen et al., 2015). Here, the 2-MGA/MTL
ratios in autumn and winter exceeded those observed in spring and summer (Figure 7b), indicating that
elevated $NO_x$ concentrations in the autumn and winter may enhance the formation of 2-MGA. In MTLs, 2-
methylthreitol and 2-methylerythritol (MTL1 and MTL2) exhibited a significant linear correlation ($R^2 = 0.68$,
$P < 0.01$), indicating that they may originate from similar sources and/or share similar formation pathways.
The ratio of MTL2 to MTLs (MTL2/MTLs) can reflect the transformation pathways of MTLs. The ratio of
MTL2/MTLs = 0.35, 0.61, 0.76, and 0.90 corresponded to isoprene secondary transformation from biogenic
sources, OH-rich conditions (low $NO_x$), $NO_x$-rich conditions, and liquid-phase oxidation by $H_2O_2$, respectively
(Kleindienst et al., 2009; Nozière et al., 2011). The results showed that the MTL2/MTLs values in spring (0.64
± 0.12) and summer (0.59 ± 0.11) were relatively low and closer to the values for biogenic sources and OH-
rich secondary transformation. However, the values in autumn and (0.66 ± 0.10) winter (0.69 ± 0.06) were
relatively high, approaching those for $NO_x$-rich secondary transformation and liquid-phase oxidation by $H_2O_2$.
This indicates that in spring and summer, MTLs were more likely derived from biogenic sources and OH-rich
secondary transformation, whereas in autumn and winter, a shift was observed toward NOx-rich secondary
transformations and liquid-phase oxidation by $H_2O_2$.

Monoterpenes represent a crucial component of BVOC emissions, accounting for approximately 11% of the

annual emissions, with an annual emission of 110 Tg C (Guenther et al., 1995). Here, we identified four





oxidation products of monoterpene (pinonic acid (PNA); pinic acid (PA); 3-methyl-1,2,3-butanetricarboxylic
acid (MBTCA); and 3-hydroxyglutaric acid (3-HGA)) at concentrations of 1.69 ± 0.99, 1.47 ± 0.86, 0.79 ±
0.61, and 0.77 ± 0.52 ng m$^{-3}$, respectively (Figure 6). The concentrations of PNA and PA exceeded those of
MBTCA and 3-HGA. PNA and PA are produced via the oxidation of α/β-pinene via reactions with $O_3$ and OH
radicals, and the α/β-pinene detected in the aerosol samples is mainly derived from biomass burning and higher
plant release (Hallquist et al., 2009; Eddingsaas et al., 2012; Zhang et al., 2015; Iyer et al., 2021). The
predominance of PNA over PA is attributable to its relatively higher vapor pressure, consistent with previous
findings (Fu et al., 2008 and 2010). PNA and PA are recognized as first-generation oxidation products of α-/β-
pinene, whereas MBTCA represents a second-generation oxidation product formed via the further
photooxidation of PNA and PA with OH radicals. The relative concentrations of such first- and second-
generation oxidation products (M/P) reflect the oxidation degree and aging status of monoterpene compounds
(Ding et al., 2014). Here, the M/P values were higher in spring (0.25 ± 0.11) and summer (0.32 ± 0.10) than in
autumn (0.16 ± 0.09) and winter (0.24 ± 0.10). This suggests that higher temperatures and/or stronger radiation
in spring and summer promoted the oxidation of monoterpene compounds (Figure 7d). The formation of 3-
HGA is considered to occur via a ring-opening mechanism, probably linked to heterogeneous reactions of
monoterpenes with irradiation in $NO_x$-rich environments (Jaoui et al., 2005; Claeys et al., 2007). The ratio of
MBTCA to 3-HGA (MBTCA/3-HGA) can be used to distinguish monoterpenes, as the secondary
transformation of α-pinene yields MBTCA at significantly higher rates relative to 3-HGA compared with β-
pinene (Jaoui et al., 2005). Here, the annual MBTCA/3-HGA value was 1.18 ± 0.65, close to those observed in
urban environments in the United States (0.81 ± 0.32) and China (0.68 ± 0.65) (Lewandowski et al., 2013;Ding
et al., 2014). The MBTCA/3-HGA value was higher in spring (1.31 ± 0.66) and summer (1.45 ± 0.70) than in
autumn (1.09 ± 0.72) and winter (0.88 ± 0.32). This suggests that the contribution of α-pinene to monoterpene
was higher in spring and summer than in autumn and winter (Figure 7e). The ratio of the total isoprene to
monoterpene oxidation products (Iso/Pine) was lower in autumn and winter than in spring and summer (Figure
7f), indicating that the higher temperatures in spring and summer favored isoprene release.
BVOC emissions include a class of sesquiterpene compounds, among which β-caryophyllene is the most
abundant and frequently reported (Ding et al., 2014; Fan et al., 2020). β-caryophyllinic acid, a product of the



ozonolysis or photooxidation of β-caryophyllene, predominantly originates from biomass burning and natural
plant emissions, including those from pine and birch trees (Helmig et al., 2006; Duhl et al., 2008). Here, β-
caryophyllinic acid was detected at a concentration of $2.74 \pm 1.92$ ng m$^{-3}$. The β-caryophyllinic acid
concentrations in autumn ($3.16 \pm 1.77$ ng m$^{-3}$) and winter ($4.63 \pm 1.61$ ng m$^{-3}$) exceeded those observed in
spring ($1.49 \pm 0.68$ ng m$^{-3}$) and summer ($1.14 \pm 0.47$ ng m$^{-3}$) (Figure 6). The higher concentrations of
caryophyllinic acid in autumn and winter may be associated with increased biomass burning and subsequent
secondary transformation of β-caryophyllene. During the sampling period, a significant positive linear
correlation was observed between β-caryophyllinic acid and levoglucosan, supporting this inference ($R^2$ =
0.66, P < 0.01).
**3.4.2 Anthropogenic secondary organic aerosol tracers**
Anthropogenic SOA tracers include hydroxy and aromatic acids, which primarily originate from the secondary
transformation of AVOCs. Although global AVOC emissions are relatively modest at 110 Tg C yr$^{-1}$ (Piccot et
al., 1992) compared with BVOC emissions, which reach 1,150 Tg C yr$^{-1}$ (Guenther et al., 1995), the
contribution of anthropogenic sources to SOA is frequently more pronounced in urban environments because
of the impact of human activities (von Schneidemesser et al., 2010; Ding et al., 2012; Li et al., 2019). AVOC
emissions in urban settings can enhance the oxidation of BVOCs, promoting SOA formation (Carlton et al.,
2010; Hoyle et al., 2011). Here, we identified two primary anthropogenic SOA tracers, 2,3-dihydroxy-4-
oxopentanoic acid (DHOPA) and phthalic acid at concentrations of $1.73 \pm 1.10$ and $6.38 \pm 4.05$ ng m$^{-3}$,
respectively (Figures 5 and 6). DHOPA and phthalic acid are recognized as important markers for
anthropogenic SOAs, produced via the oxidation of toluene and PAHs, such as naphthalene (Kawamura and
Ikushima, 1993; Kleindienst et al., 2007 and 2012). Their concentrations were significantly elevated during
autumn and winter compared with spring and summer (Figures 5 and 6). This indicated a marked increase in
the contribution of anthropogenic sources to SOA during the cold months.
**3.5 Source apportionment of organic carbon and aerosol**
Employing a tracer-based methodology, we quantified the contributions of POC and SOC to the total OC
(Figure 8). The results showed that the POC and SOC concentrations were $3.22 \pm 1.81$ and $0.50 \pm 0.32$ μg m$^{-3}$
(Figure 8a and 8b), accounting for $53.30\% \pm 12.53\%$ and $8.33\% \pm 3.29\%$ of the measured OC, respectively



(Figure 8f). These results were within the reported ranges of POC (5% to 76%) and SOC (3% to 56%)
proportions reported in other studies (Stone et al., 2009; Guo et al., 2012; Fan et al., 2020; Xu et al., 2021;
Haque et al., 2023). Anthropogenic sources, including biomass burning, coal combustion, motor vehicle
emissions, and cooking, contributed the majority of POC ($2.89 \pm 1.72$ µg m$^{-3}$, accounting for $88.98\% \pm$
$6.24\%$). Natural sources, such as plant debris and fungal spores, contributed relatively little ($0.33 \pm 0.17$ µg
m$^{-3}$, accounting for $11.01\% \pm 6.24\%$). For SOC, anthropogenic contributions ($0.33 \pm 0.25$ µg m$^{-3}$, accounting
for $60.67\% \pm 21.29\%$) exceeded biogenic contributions ($0.17 \pm 0.09$ µg m$^{-3}$, accounting for $39.05\% \pm$
$21.15\%$). The concentrations and relative abundances of anthropogenic POC and SOC were higher in autumn
and winter, whereas biogenic POC and SOC exhibited greater concentrations and relative abundances during
spring and summer (Figures 8a, 8b, 8d, and 8e). These results underscore the significant influence of biogenic
emissions during the warm months, whereas anthropogenic emissions exert a pronounced effect during the
cold months. This situation is primarily attributed to the elevated temperatures, and increased solar radiation
during the warm months facilitates VOC release from vegetation, thereby promoting biogenic POC emissions
and SOC formation. Contrarily, the rise in anthropogenic POC emissions from biomass burning and coal
combustion during the cold months fosters the development of anthropogenic SOC in the atmosphere.
Similarly, using a tracer-based methodology, we quantified the contributions of POA and SOA to PM$_{2.5}$ (Figure
S4). The findings indicated that POA and SOA contributed $6.13 \pm 3.58$ and $1.02 \pm 0.63$ µg m$^{-3}$, accounting for
$21.02\% \pm 6.50\%$ and $3.47\% \pm 1.41\%$ of the observed PM$_{2.5}$ mass in Nanchang, respectively. The
anthropogenic POA ($5.50 \pm 3.41$ µg m$^{-3}$, accounting for $88.74\% \pm 6.46\%$) and SOA ($0.65 \pm 0.48$ µg m$^{-3}$,
accounting for $58.79\% \pm 21.30\%$) exceeded the natural POA ($0.63 \pm 0.33$ µg m$^{-3}$, accounting for $11.26\% \pm$
$6.46\%$) and SOA ($0.37 \pm 0.20$ µg m$^{-3}$, accounting for $40.93\% \pm 21.19\%$), respectively. The POA and SOA
exhibited patterns analogous to those of POC and SOC, with greater contributions recorded in autumn and
winter compared with spring and summer. The tracer-based method inherently bears a degree of uncertainty,
primarily due to the variability in the mass fractions of representative tracers in OC and OA ($f_{oc}$ and $f_{oa}$) in
different observational studies. Nevertheless, this method has been widely employed to estimate the
contributions of various primary and secondary sources to OC and aerosol, with relatively reasonable results.
Additional observational studies to determine the representative tracers in emission sources and their mass





fractions in OC and OA are crucial to reducing this uncertainty (Oros and Simoneit, 2000; He et al., 2004;
Zhao et al., 2007; Zhang et al., 2008; Kleindienst et al., 2012; Andreae, 2019).

We employed the CMB model to calculate the contributions of different sources of POC and SOC to OC.

The results derived from the CMB approach were largely consistent with those obtained using the tracer-based
method (Figure S5). The CMB model operates on principles analogous to the tracer-based method, relying on
the mass fractions of characteristic tracers in OC and OAs from emission sources to ascertain the contributions
of different sources (Stone et al., 2009). Thus, theoretically, the results generated by the CMB model should be
consistent with those obtained from the tracer-based method. The present results confirmed that both methods
exhibited similar reliabilities. We employed the EC-based method to estimate the total POC and SOC
concentrations. Dissimilar to the tracer-based method, which quantifies partial POC and SOC concentrations
from specific sources, the EC-based method simply partitions the OC into POC and SOC, which calculates the
total POC and SOC concentrations. The results demonstrated that the overall trends of the total POC and SOC
calculated by the EC-based method were consistent with those obtained via the tracer-based method (Figure
S6).
**3.6 Characteristics of organic aerosols during winter pollution**
From an annual timescale perspective, anthropogenic POC and SOC are synchronous with the measured OC
and observed $PM_{2.5}$ mass (Figure S6). Correlation analysis revealed that anthropogenic POC and SOC
exhibited a significant positive correlation (r = 0.72–0.80, P < 0.01) with the measured OC (Figure 9a).
Redundancy analysis revealed that anthropogenic POC and SOC significantly contributed (40%–65%, P <
0.01) to the variations in OC concentration (Figure 9a). This suggests that anthropogenic OC and SOC are
important factors influencing changes in the OC and $PM_{2.5}$ mass throughout the sampling period. Variations in
POC, SOC, OC, and $PM_{2.5}$ mass were associated with specific meteorological conditions and gaseous pollutant
concentrations. For instance, high POC, SOC, OC, and $PM_{2.5}$ concentrations corresponded to elevated
atmospheric pressure, reduced precipitation, and increased $NO_2$ concentrations (Figure S2). This suggests that
meteorological conditions and gaseous pollutant concentrations could also influence POC, SOC, OC, and
$PM_{2.5}$ mass.

However, winter presents a different scenario. During the entire winter pollution period and separate PM



pollution episodes, POC did not maintain good consistency with the changes in OC and PM$_{2.5}$ mass, whereas
SOC remained synchronized with their variations (Figure S6). Correlation and redundancy analyses further
revealed that source-specific SOC, derived using the tracer methods, and the total SOC, calculated using the
EC-based method, exhibited significant positive correlations with the measured OC (r > 0.6, P < 0.05) and
significantly contributed (>40%, P < 0.05) to their variations (Figures 9b–9k). These findings indicate that on
shorter timescales, particularly during brief PM pollution episodes lasting several days, SOC was a critical
factor influencing the OC and PM$_{2.5}$ mass. The increased SOC concentration may be associated with elevated
temperatures and NO$_x$ concentrations during winter pollution episodes. This inference was supported by a
significant linear positive correlation between temperature, NO$_2$ concentrations, and SOC concentrations
observed during several winter pollution episodes (Figure 10). An increase in short-term solar radiation
intensity and oxidant levels have been shown to accelerate SOC formation (Fry et al., 2009; Ng et al., 2017; Li
et al., 2018; Ren et al., 2019). Nevertheless, the contribution of POC should not be underestimated, as its levels
remained relatively high throughout the winter. This elevated POC concentration can also promote SOC
transformation (Weber et al., 2007; Carlton et al., 2010; Hoyle et al., 2011; Srivastava et al., 2022).
**4 Conclusions**
We investigated the composition and concentration of major polar organic compounds in PM$_{2.5}$ samples
collected over a year in Nanchang, Central China. The results revealed relatively high concentrations of FAs,
fatty alcohols, and saccharides, whereas lignin, resin products, sterols, glycerol, hydroxy acids, and aromatic
acids were detected at low levels.

The study findings indicate that the organic components in the PM$_{2.5}$ of Nanchang are predominantly

derived from anthropogenic and natural sources. Anthropogenic sources were the primary contributors to OC
and OA, and primary sources contributed more than secondary sources. Throughout the sampling period, we
observed that the anthropogenic contributions significantly increased during autumn and winter, whereas
biogenic contributions increased in spring and summer.

This study highlights the critical role of anthropogenic POC and SOC in influencing atmospheric PM$_{2.5}$

pollution over an annual sampling period. During short-term pollution episodes in winter (lasting several
days), the rapid secondary transformation emerged as the primary driver of OC increment and PM$_{2.5}$ pollution.





Elevated primary emissions and favorable oxidation conditions, such as increased light intensity and $NO_x$
levels, were identified as key factors facilitating the rapid secondary transformation of OC during the winter
pollution episodes.

The study findings underscore the necessity for targeted management strategies that consider primary and

secondary anthropogenic emission sources across different seasons and pollution periods. Although the tracer-
based methodology provided preliminary insights into the OC and OA from diverse sources, the study
encountered inherent limitations in characteristic compound identification. The source apportionment analysis
potentially underestimated contributions from unidentified primary and secondary sources because of the
restricted range of tracers used. To address this, future research should prioritize comprehensive observational
studies aimed at identifying representative molecular tracers across a broader spectrum of emissions and
quantifying the precise mass fractions of such tracers in OC and OAs.

**Data availability.** The dataset for this paper is available upon request from the corresponding author
(xiaohuayun@sjtu.edu.cn).

**Competing interests.** The authors declare that they have no conflict of interest.

**Author contributions.** Huayun Xiao conceptualized and designed the research. Sampling was conducted by
Zicong Li, while laboratory analyses were carried out by Wei Guo, Zicong Li, and Renguo Zhu. Data
interpretation was supported by Zhongkui Zhou and Hongwei Xiao. The manuscript was primarily written by
Wei Guo, with contributions and consultations from all other authors.

**Acknowledgements.** We thank Ziyue Zhang and Liqin Cheng for their help with sampling and laboratory
work. This study was supported by the National Natural Science Foundation of China (grant number
41863002); the Jiangxi Provincial Natural Science Foundation (grant number 20242BAB25193); the Open
Foundation of Jiangxi Province Key Laboratory of the Causes and Control of Atmospheric Pollution, East



China University of Technology (grant number AE2209); and the Open Foundation of Yunnan Province Key
Laboratory of Earth System Science, Yunnan University (grant number ESS2024002).

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

Background Atmosphere of China, Environ. Sci. Technol., 58, 1244–1254,
https://doi.org/10.1021/acs.est.3c08491, 2024.
Zhang, Y. X., Schauer, J. J., Zhang, Y. H., Zeng, L. M., Wei, Y. J., Liu, Y., and Shao, M.: Characteristics of
Particulate Carbon Emissions from Real-World Chinese Coal Combustion, Environ. Sci. Technol., 42,
5068–5073, https://doi.org/10.1021/es7022576, 2008.
Zhang, Y. L., Kawamura, K., Agrios, K., Lee, M., Salazar, G., and Szidat, S.: Fossil and Nonfossil Sources of
Organic and Elemental Carbon Aerosols in the Outflow from Northeast China, Environ. Sci. Technol., 50,
6284–6292, https://doi.org/10.1021/acs.est.6b00351, 2016.
Zhao, Y. L., Hu, M., Slanina, S., and Zhang, Y. H.: Chemical Compositions of Fine Particulate Organic Matter
Emitted from Chinese Cooking, Environ. Sci. Technol., 41, 99–105, https://doi.org/10.1021/es0614518,
921      2007.

Zhu, C. M., Kawamura, K., and Fu, P. Q.: Seasonal variations of biogenic secondary organic aerosol tracers in
Cape Hedo, Okinawa, Atmos. Environ., 130, 113–119, https://doi.org/10.1016/j.atmosenv.2015.08.069,
924      2016.

Zhu, R. G., Xiao, H. Y., Cheng, L., Zhu, H., Xiao, H., and Gong, Y. Y.: Measurement report: Characterization
of sugars and amino acids in atmospheric fine particulates and their relationship to local primary sources,
Atmos. Chem. Phys., 22, 14019–14036, 10.5194/acp-22-14019-2022, 2022.











**Figure Captions**

Figure 1. (a) Average $PM_{2.5}$ concentrations across various cities in China during winter 2023–2024 (December 2023 to February 2024). (b) Location of sampling sites in Nanchang.

Figure 2. One-year time series of organic carbon (OC), elemental carbon (EC), OC/EC ratios, $PM_{2.5}$ concentrations, and concentrations of polar organic compounds.

Figure 3. Molecular characteristics and seasonal variations of the major polar compounds in $PM_{2.5}$.

Figure 4. The concentration ratios and its seasonal variations of the major polar compounds. Each box plot illustrates the mean (centerline), interquartile range (box encompassing the 25th to 75th percentiles), and standard deviation (whiskers).

Figure 5. Molecular characteristics and seasonal variations of the minor polar compounds in $PM_{2.5}$.

Figure 6. Molecular characteristics and seasonal variations of the SOA tracers in $PM_{2.5}$.

Figure 7. The concentration ratios and its seasonal variations of the SOA tracers. Box plots represent the mean (centerline), interquartile range (box encompassing the 25th to 75th percentiles), and standard deviation (whiskers).

Figure 8. Concentrations and relative abundances of primary organic carbon (POC) from biomass burning ($POC_{bb}$), coal combustion ($POC_{cc}$), vehicle exhaust ($POC_{ve}$), cooking ($POC_{c}$), plant debris ($POC_{pd}$), and fungal spores ($POC_{fs}$), alongside secondary organic carbon (SOC) from isoprene ($SOC_{i}$), α/β-pinene ($SOC_{p}$), β-caryophyllene ($SOC_{c}$), toluene ($SOC_{t}$), and naphthalene ($SOC_{n}$). POC and SOC concentrations were estimated by the tracer-based method, and other OC (OOC) was obtained by subtracting estimated OC from measured OC.

Figure 9. Pearson correlation between POC, SOC, and the measured OC during the annual sampling period and winter pollution episodes. Circle size represents the contribution (%) of POC and SOC to the variation in measured OC, as determined by redundancy analysis. Magenta circles represent P-values from correlation and redundancy analyses that are less than 0.05, while gray circles indicate P-values greater than 0.05. Definitions: $POC_{bb}$ (biomass burning), $POC_{cc}$ (coal combustion), $POC_{ve}$ (vehicle exhaust), $POC_{c}$ (cooking), $POC_{pd}$ (plant debris), $POC_{fs}$ (fungal spores), and $POC_{EC-based}$ (based on EC method). $SOC_{i}$, $SOC_{p}$, $SOC_{c}$, $SOC_{t}$, $SOC_{n}$, and



$SOC_{EC\text{-based}}$ refer to SOC from isoprene, α/β-pinene, β-caryophyllene, toluene, naphthalene, and EC-based
methods, respectively.
Figure 10. Linear correlation between temperature, $NO_2$, and SOC during specific winter pollution episode.
SOC concentrations were estimated by the tracer-based method.


















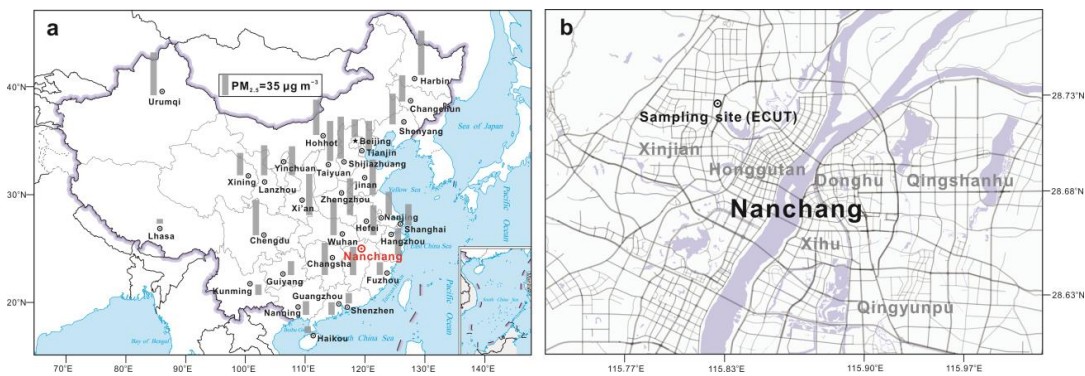

Figure 1. (a) Average PM$_{2.5}$ concentrations across various cities in China during winter 2023–2024 (December

2023 to February 2024). (b) Location of sampling sites in Nanchang.



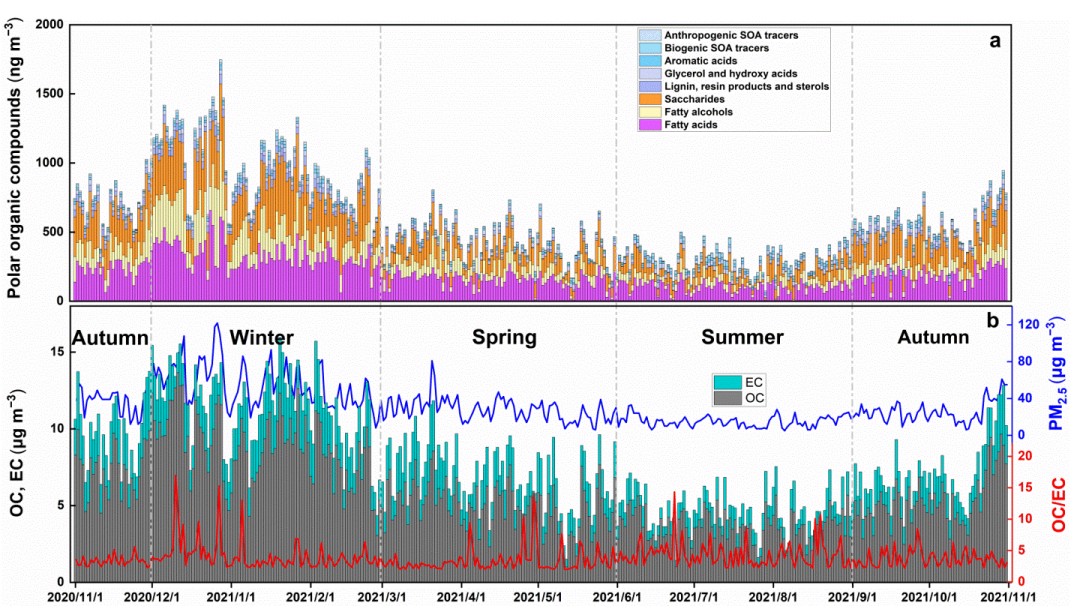

Figure 2. One-year time series of organic carbon (OC), elemental carbon (EC), OC/EC ratios, PM$_{2.5}$ concentrations, and concentrations of polar organic compounds.



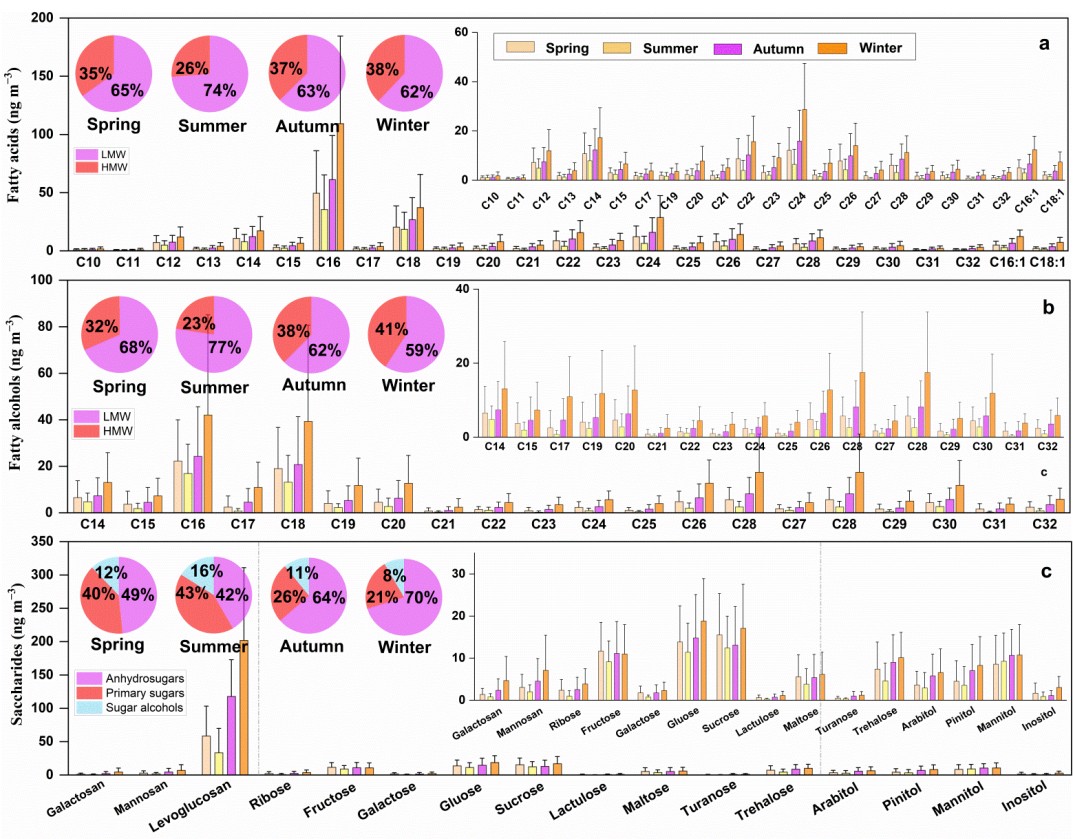


Figure 3. Molecular characteristics and seasonal variations of the major polar compounds in PM$_{2.5}$.












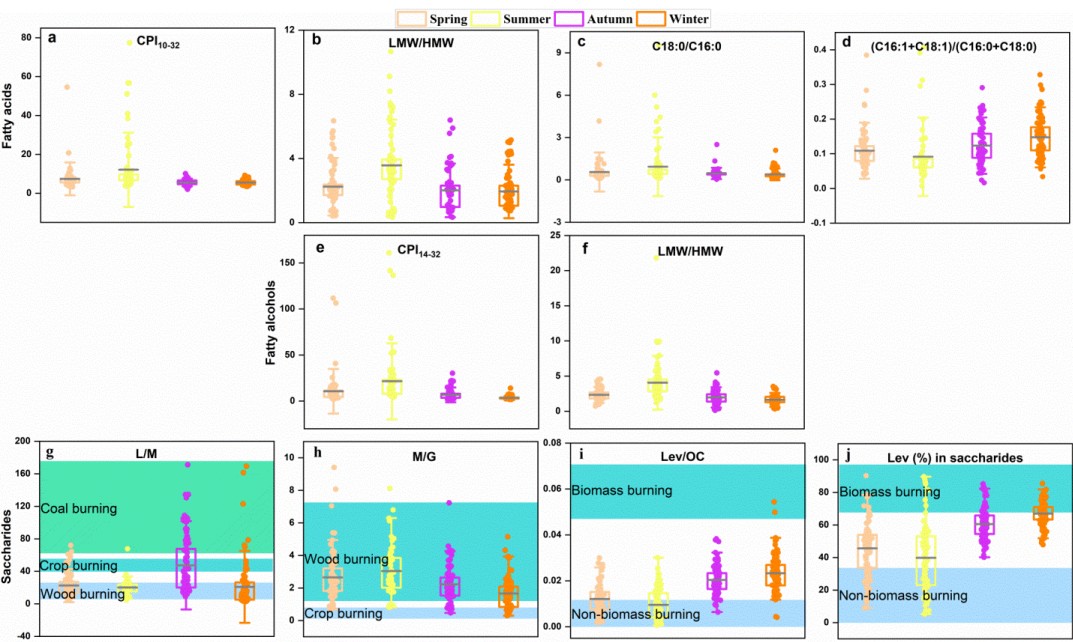


Figure 4. The concentration ratios and its seasonal variations of the major polar compounds. Each box plot

illustrates the mean (centerline), interquartile range (box encompassing the 25th to 75th percentiles), and

standard deviation (whiskers).




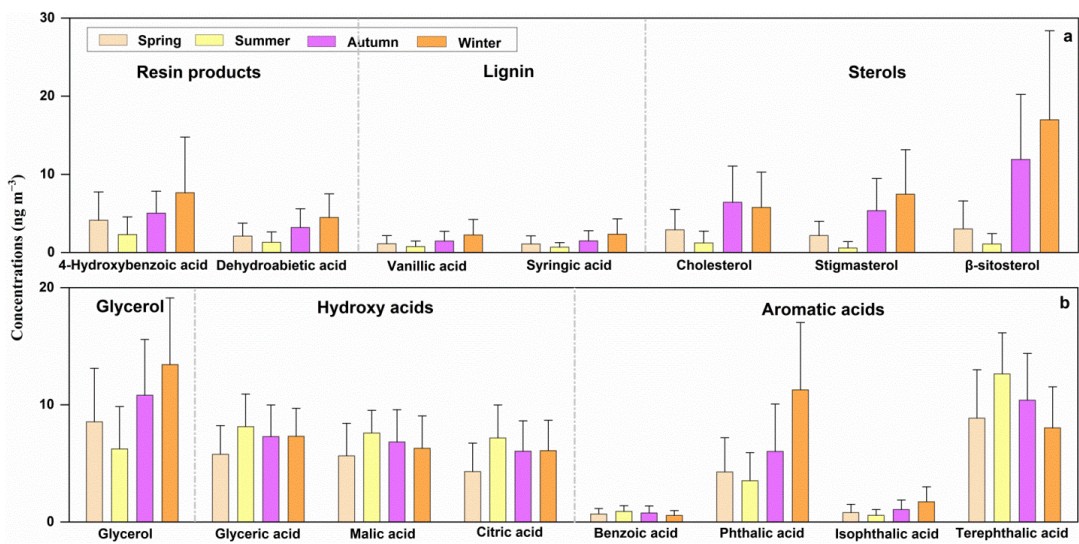

Figure 5. Molecular characteristics and seasonal variations of the minor polar compounds in $PM_{2.5}$.



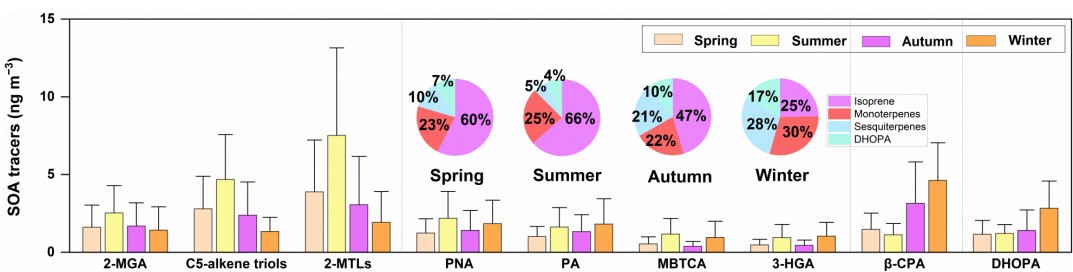

Figure 6. Molecular characteristics and seasonal variations of the SOA tracers in PM$_{2.5}$.



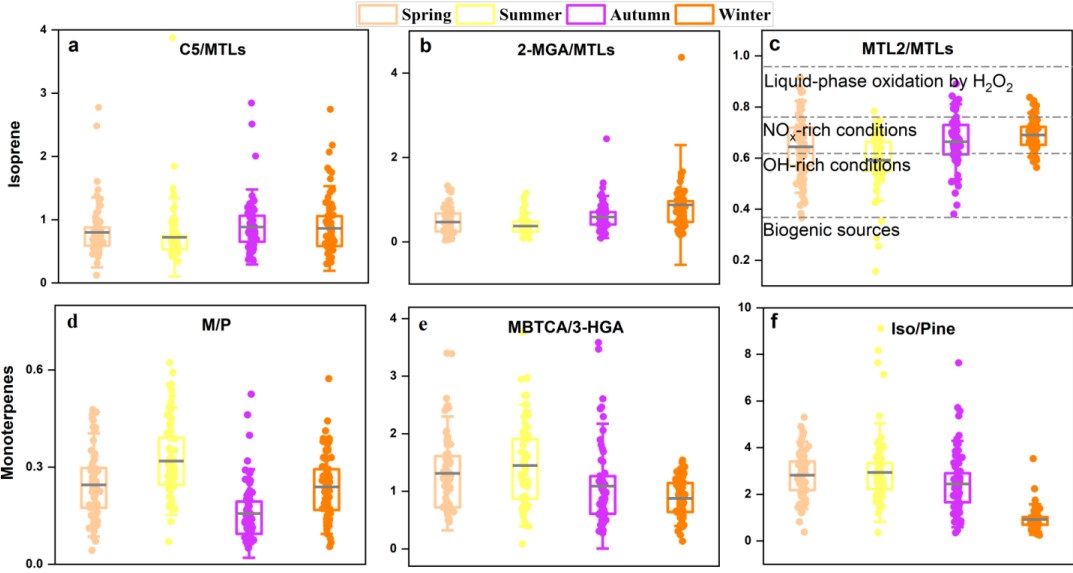

Figure 7. The concentration ratios and its seasonal variations of the SOA tracers. Box plots represent the mean (centerline), interquartile range (box encompassing the 25th to 75th percentiles), and standard deviation (whiskers).



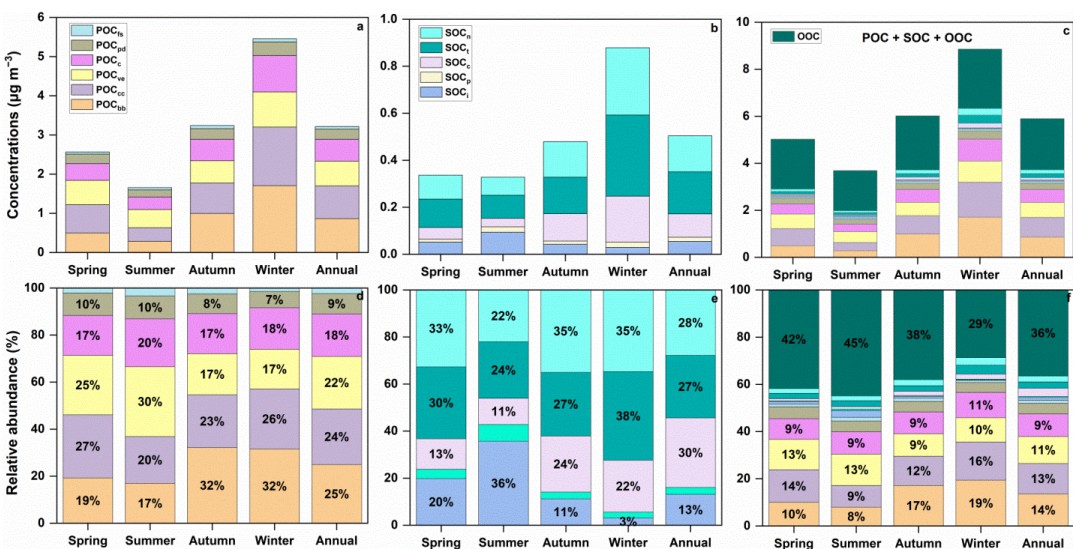

Figure 8. Concentrations and relative abundances of primary organic carbon (POC) from biomass burning (POC$_{bb}$), coal combustion (POC$_{cc}$), vehicle exhaust (POC$_{ve}$), cooking (POC$_{c}$), plant debris (POC$_{pd}$), and fungal spores (POC$_{fs}$), alongside secondary organic carbon (SOC) from isoprene (SOC$_{i}$), α/β-pinene (SOC$_{p}$), β-caryophyllene (SOC$_{c}$), toluene (SOC$_{t}$), and naphthalene (SOC$_{n}$). POC and SOC concentrations were estimated by the tracer-based method, and other OC (OOC) was obtained by subtracting estimated OC from measured OC.

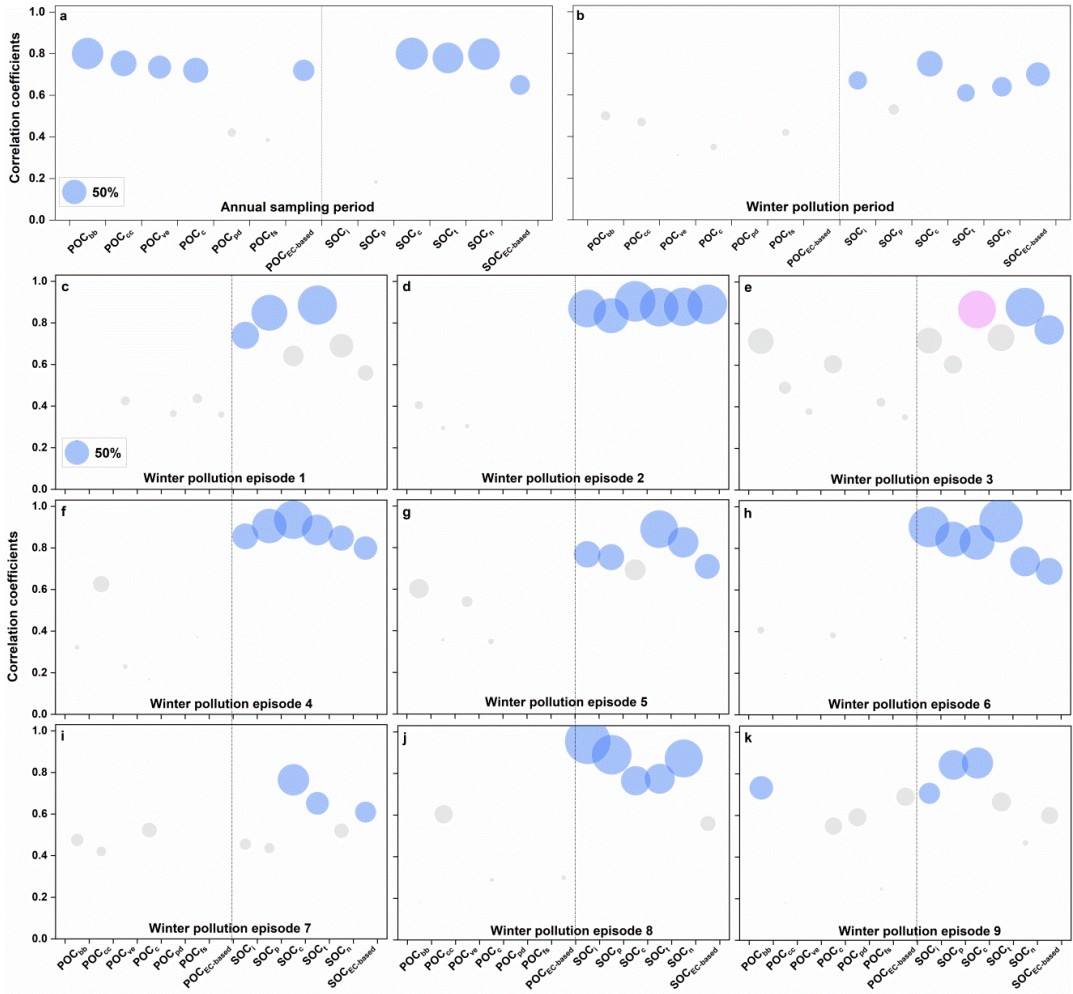

Figure 9. Pearson correlation between POC, SOC, and the measured OC during the annual sampling period and winter pollution episodes. Circle size represents the contribution (%) of POC and SOC to the variation in measured OC, as determined by redundancy analysis. Magenta circles represent P-values from correlation and redundancy analyses that are less than 0.05, while gray circles indicate P-values greater than 0.05. Definitions: $POC_{bb}$ (biomass burning), $POC_{cc}$ (coal combustion), $POC_{ve}$ (vehicle exhaust), $POC_c$ (cooking), $POC_{pd}$ (plant debris), $POC_{fs}$ (fungal spores), and $POC_{EC\text{-}based}$ (based on EC method). $SOC_i$, $SOC_p$, $SOC_c$, $SOC_t$, $SOC_n$, and $SOC_{EC\text{-}based}$ refer to SOC from isoprene, α/β-pinene, β-caryophyllene, toluene, naphthalene, and EC-based methods, respectively.



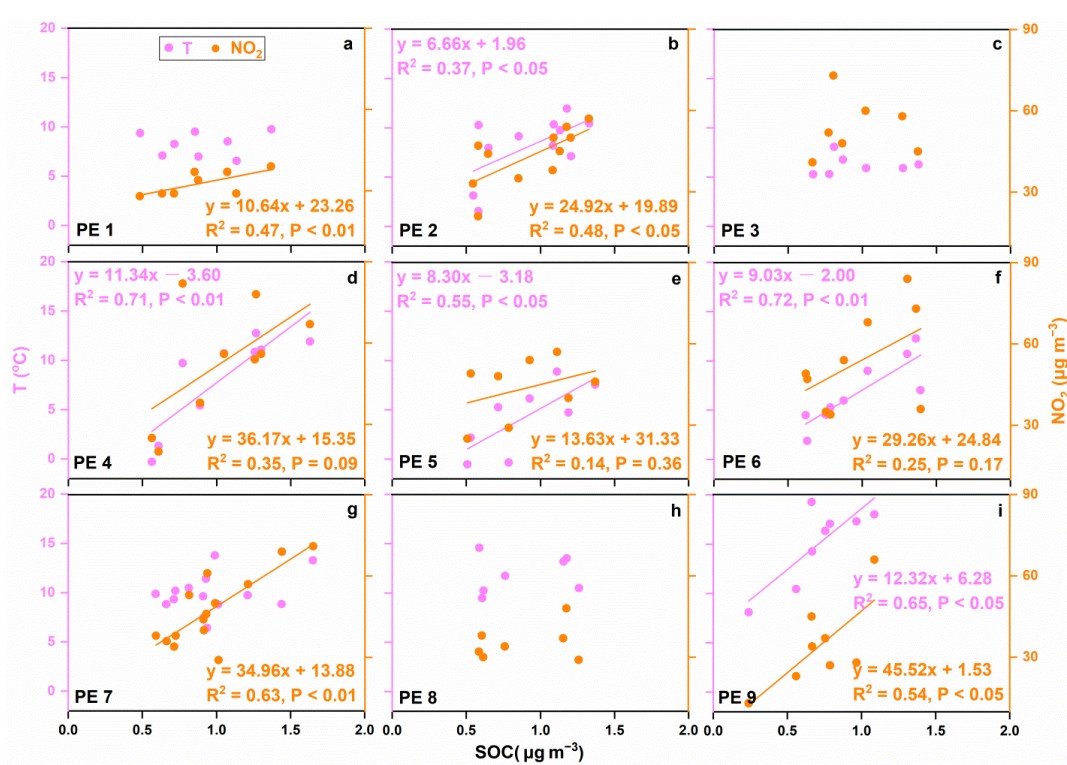

Figure 10. Linear correlation between temperature, $NO_2$, and SOC during specific winter pollution episode.

SOC concentrations were estimated by the tracer-based method.