# Peer review of "Contributions of primary anthropogenic sources and rapid secondary"

_EGUsphere, 2024_

## Author Comment (AC1)

**Response to the editor's and reviewers' comments point-by-point**

**Editor's comments:**

**General comments.** Figure 1a may contain a territory that is disputed according to the United Nations. If and when the manuscript is accepted for final revised publication, you will be asked to choose one of the following options: (a) you could remove the disputed territory from the map and submit new figure files, or (b) we could add a statement that some figures contain disputed territories.

**Response:** Thank you for your guidance regarding Figure 1a. We have revised Figure 1a as requested.

We should also point out that in response to Reviewer 2's comment that "Figure 1a did not present the original results and mainly supports the introduction, and Figure 1b provided minimal information and could be completely removed," we have revised Figure 1 as follows: Figure 1b has been removed, and Figure 1a has been relocated to the Supplementary Materials and renumbered as Figure S1c.

Furthermore, to better support the introduction, Figure S1 in the Supplementary Materials has been expanded and now includes: (a) annual average $PM_{2.5}$ concentration trends in China and Nanchang from 2013 to 2023 (Figure S1a), and (b) daily $PM_{2.5}$ concentration variations during the winter of 2023–2024 (Figure S1b).

These changes are detailed in Figure S1 of the Supplementary Materials.

**Reviewer 1:**

**General comments.** In this article, the authors present one year of daily speciated organic aerosol measurements taken in Nanchang, China. They collected filters using high-volume samplers which were analyzed offline. The authors then performed source apportionment which revealed large contributions of primary anthropogenic sources. The authors describe the behavior of each component class, describing correlations between species, concentration ratios and the influence of seasonality. The largest concentrations were observed in the winter and autumn seasons, driven by coal combustion and biomass burning although individual pollution episodes were correlated with increased SOC indicating implying fast processing. While a minor component, biogenic aerosols contributed a larger fraction of the aerosol in the summertime.

The content of the article and the scientific analysis is well within the scope of ACP. Although the data analysis methods are not novel, the article adds to compounding evidence of anthropogenic POA and SOA contributing to wintertime pollution events. I believe the conclusions are consistent with the data obtained and methods described in previous work.

Structurally the paper is sound, the abstract is a good summary of the contents, the authors cite

proper and relevant references, and the language is fluid and easy to read. I believe the article can be published in ACP after minor revisions:.

**Specific comments 1.** Figure 2 I believe $PM_{2.5}$ and OC/EC in panel b can be separated into a panel c to aid clarity.

**Response:** Thank you very much for your suggestion. In the revised manuscript, we have separated the $PM_{2.5}$ and OC/EC data from panel b into new panels a and b, respectively, to improve the readability and clarity of the figures. The original panel a has been renumbered as panel c, and Figure 2 has been renumbered as Figure 1.

These changes are detailed in Figure 1 of the revised manuscript.

**Specific comments 2.** Figure 4 and 7 the yellow markers on a white background are difficult to see. Changing them to red or adding a black border would improve readability.

**Response:** Thank you for your comment regarding Figures 4 and 7. In the revised manuscript, the yellow markers in these figures have been changed to red to improve readability. Additionally, Figures 4 and 7 have been relocated to the Supplementary Materials and renumbered as Figures S3 and S5.

**Specific comments 3.** The wintertime pollution events are under analyzed. While the authors described the correlations observed for SOC in Figures 9 and 10 there is no comparison between the overall composition of the aerosol during these events and the rest of the winter and the year overall. Figure 8 shows wintertime concentrations are driven by POC but does not show the overall contributions during pollution events, SOC being a minor component in the average wintertime data.

**Response:** Thank you very much for your valuable comments. We have carefully addressed your concerns regarding the under-analysis of wintertime pollution events. In the revised manuscript, we have conducted a more detailed analysis of organic aerosol composition and source contributions specifically during winter pollution events and compared these to the remainder of the winter and the full sampling period. The key revisions include:

1. Expanded Analysis:

We performed comparative analyses of organic composition, pollution characteristics, and source contributions for: (a) the entire sampling period, (b) winter pollution periods, (c) the remainder of the winter season, and (d) individual pollution episodes.

2. Figure Revisions:

In Figure 9 (now Figure 6 in the revised manuscript), we include comparisons of SOC correlations with source contributions across the above periods.

Figure 10 (now Figure 7) presents linear correlations between SOC and both temperature and $NO_2$

for these specific timeframes.

Figure 8 (now Figure 5) provides comparisons of POC and SOC contributions from different sources during winter pollution periods versus the rest of winter.

A new Supplementary Figure S6 shows source contributions of POC and SOC during individual winter pollution episodes.

3. Expanded Discussion:

We expanded the discussion in Section 3.6, "Characteristics of organic aerosols during winter pollution," to include these comparative correlation analyses.

For example: "However, winter presents a different scenario. Regardless of whether it is during the winter pollution periods or other times in winter, POC did not maintain good consistency with the changes in the total OC content and $PM_{2.5}$ mass, whereas SOC remained synchronized with their variations (Figure S9). This phenomenon is particularly pronounced during individual pollution episodes in winter (Figure S9). Correlation and redundancy analyses further revealed that source-specific SOC, derived using the tracer methods, and the total SOC, calculated using the EC-based method, exhibited significant positive correlations with the measured OC ($r > 0.6$, $P < 0.05$) and significantly contributed ($> 40\%$, $P < 0.05$) to their variations in entire winter pollution periods and individual pollution episodes (Figures 6b and 6d–6l). Given the relatively small number of nonpolluted days in winter and their scattered distribution across different dates, the correlations and contribution levels were not as evident during the remainder of the winter season (Figures 6c). These findings indicate that on shorter timescales, particularly during brief PM pollution episodes lasting several days, SOC was a critical factor influencing the total OC content and $PM_{2.5}$ mass. " **(lines 555–566, pages 21–22)**

"The increased SOC concentration may be associated with elevated temperatures and $NO_x$ concentrations during winter pollution episodes. This inference was supported by a significant linear positive correlation between temperature, $NO_2$ concentrations, and SOC concentrations observed during entire winter pollution periods and individual pollution episodes (Figures 7b and 7d–7l). An increase in short-term solar radiation intensity and oxidant levels have been shown to accelerate SOC formation (Fry et al., 2009; Ng et al., 2017; Li et al., 2018; Ren et al., 2019). Nevertheless, the contribution of POC should not be underestimated as its levels remained relatively high throughout the winter. This elevated POC concentration can also promote SOC transformation (Weber et al., 2007; Carlton et al., 2010; Hoyle et al., 2011; Srivastava et al., 2022). Although a negative correlation between temperature and SOC concentrations is observed on an annual timescale (Figure 7a), high $NO_2$ levels are also associated with high SOC concentrations over longer periods. The negative correlation between temperature and SOC concentrations on an annual scale mainly arises because emission sources and SOC transformation products decrease during warmer seasons, whereas they increase in colder seasons, especially winter (Haque et al., 2019; Fan et al., 2020). Outside pollution periods in winter, variations in temperature and $NO_2$ concentrations are minimal, which limits their promoting effect on SOC concentration increases (Figure 7c)." **(lines 567–581, page 22)**

We also expanded the discussion in Section 3.5, "Source apportionment of organic carbon and aerosols" to include these comparative analyses.

For example: "During winter, including both pollution episodes (Winter-P) and the remainder period (Winter-R), the contribution proportions of two anthropogenic POC sources (biomass and coal combustion, 34%–36%) and anthropogenic SOC precursors (toluene and naphthalene, 6%–7%) were consistently higher than the annual averages (27% for these two anthropogenic POC and 5% for anthropogenic SOC) (Figure 5f). This trend was even more pronounced during individual pollution episodes (Figure S6), with anthropogenic POC contributions reaching 31%–42% and anthropogenic SOC contributions increasing to 6%–9% (Figure S6f). These findings suggest that POC from coal and biomass combustion, along with SOC originating from toluene and naphthalene, may be key drivers of wintertime organic aerosol pollution." **(lines 512–520, page 20)**

As the reviewer noted, "Figure 8 shows wintertime concentrations are driven by POC but does not show the overall contributions during pollution events, SOC being a minor component in the average wintertime data." We have addressed this point with the following explanation and discussion:

"Although the contribution of SOC to OC is substantially smaller than that of POC, the impact of SOC should not be overlooked. First, this study identified only five types of SOC products, and there may be other SOC products—such as liquid-phase SOC products and primary aging SOC products—that cannot be effectively detected using the SOC tracer method (Ding et al., 2014). Therefore, the SOC contribution estimated by the SOC tracer method is likely underestimated. Second, numerous studies have demonstrated that even a small increase in the proportion of SOC during winter pollution periods can worsen OA pollution (Li et al., 2019; Xu et al., 2021; Haque et al., 2023)." **(lines 520–526, page 20)**

For the related content, please see Section 3.5, "Source apportionment of organic carbon and aerosols" **(lines 491–542, pages 19–21)**, Section 3.6, "Characteristics of organic aerosols during winter pollution" **(lines 543–587, pages 21–22)**, Figures 5–7, and Figure S6 in the revised manuscript.

**Specific comments 4.** Is there a meteorological component to the observed wintertime pollution events? Temperature is correlated with SOC but other details such as windspeed, RH etc. are not mentioned.

**Response:** Thank you for these valuable comments. We have added Figure S10 in the Supplementary Materials to illustrate the linear correlations between additional meteorological factors—such as wind speed and relative humidity—and SOC during the entire sampling period, the winter pollution period, the remainder of the winter season, and individual pollution episodes. Correspondingly, we have supplemented the discussion in Section 3.6, "Characteristics of organic aerosols during winter pollution."

Specifically: "Additionally, we analyzed the linear correlations between other meteorological

factors and SOC. In particular, we examined the relations between wind speed, relative humidity, and SOC (Figure S10). The results indicated that no clear relation existed between wind speed and SOC. Although relative humidity showed some linear association with SOC during a few isolated pollution episodes, this relation was weak, with relatively large P-values and lacking statistical significance. These findings suggest that temperature and $NO_2$ concentrations are likely the main meteorological and oxidative factors driving the increase of SOC during winter pollution periods in the Nanchang region."

For the related content, please refer to Figure S10 and Section 3.6, "Characteristics of organic aerosols during winter pollution" **(Lines 581–587, page 22)**.

**Reviewer 2:**

**General comments.** This study presents a comprehensive analysis of organic aerosol (OA) sources and their seasonal variations in PM$_{2.5}$ in Nanchang, China, using an integrated approach combining EC-based, tracer-based, and CMB (chemical mass balance) modeling methods. The research makes valuable contributions by quantifying both primary (POC/POA) and secondary (SOC/SOA) organic aerosol fractions, and it highlights the dominance of anthropogenic sources (accounting for 89% of POC/POA), particularly from biomass burning and coal combustion. The observation of rapid SOC formation during winter pollution episodes provides meaningful insights into secondary aerosol chemistry under specific meteorological conditions.

Overall, the study offers strong novelty and scientific value. However, it requires revisions to improve language clarity, methodological transparency, and analytical rigor. I recommend publication after addressing the following comments:.

**Specific comments 1.** Methodology: A clearer discussion is needed regarding the uncertainties associated with the tracer-based method and the validation of the CMB model. The authors should elaborate on the sources of uncertainty and suggest possible improvements.

**Response:** Thank you very much for your valuable comments. We have addressed these issues in Section 2.3, "Source apportionment methods," as follows:

"We employed the CMB model (version 8.2) provided by the US Environmental Protection Agency to verify the results calculated using a tracer-based method. This model assumes a CMB between the emission sources and environmental receptors. Thus, the mass of pollutants is not lost during transport from the source to the receptor, and the measured chemical component concentration at the receptor is the linear superposition of the contribution of each source class to the concentration (Stone et al., 2009). The CMB model operates on principles analogous to those involved in the tracer-based method, relying on the mass fractions of characteristic tracers in OC and OAs from emission sources to ascertain the contributions of different sources. Thus, theoretically, the results generated by the CMB model should be consistent with those obtained from the tracer-based method. The tracer-based method and CMB model inherently involve a degree of uncertainty, primarily owing to variability in the mass fractions of representative tracers in OC and OA ($f_{oc}$ and $f_{oa}$) from the same emission source across different observational studies

(Ding et al., 2012 and 2014). For instance, levoglucosan—a molecular marker for biomass burning—has been found to represent varying proportions of OC in different studies. Andreae et al. (2009) reported a levoglucosan-to-OC ratio ($f_{oc}$) of approximately 0.1186, with Zheng et al. (2002) observing a higher average value of 0.1258, and Zhang et al. (2007) reported a lower $f_{oc}$ value of 0.082 for biomass burning emissions. Nevertheless, this method has been widely employed to estimate the contributions of various primary and secondary sources to OC and aerosols, yielding relatively reasonable results (Kleindienst et al., 2007 and 2012; Stone et al., 2009; Ding et al., 2012 and 2014; Al-Naiema et al., 2017; Ren et al., 2021; Xu et al., 2021; Haque et al., 2023). To further reduce uncertainty, additional observational studies are essential for identifying representative tracers in emission sources and determining their mass fractions in OC and OA (Oros and Simoneit, 2000; He et al., 2004; Zhao et al., 2007; Zhang et al., 2008; Kleindienst et al., 2012; Andreae, 2019)."

For the related content, please see Section 2.3, "Source apportionment methods" **(Lines 137–157, page 6)** in the revised manuscript**.**

**Specific comments 2.** Data Presentation: Figures could be improved with clearer seasonal distinctions and inclusion of statistical indicators (e.g., standard deviation, p-values, and confidence intervals). This would enhance the interpretability of seasonal trends and comparisons.

**Response:** Thank you very much for your comments. We agree that enhancing the clarity of seasonal distinctions and including appropriate statistical indicators will improve the interpretability of our results. In the revised manuscript, we have updated Figures 3–7 (now renumbered as Figures 2–4, S3, and S5) to include standard deviations and to annotate significant seasonal differences with different letters. We used the Tukey test to assess significance at the 95% confidence level.

For example: "Figure 2. Molecular characteristics and seasonal variations of the major polar compounds in $PM_{2.5}$. The bars in the figure show the average concentrations of different compounds for each season, while the caps on the bars represent the standard deviations of these averages. The letters a, b, c, and d above the caps indicate whether the average concentrations differ significantly between seasons. Data points that do not share the same letter are significantly different, whereas those with the same letter are not significantly different, at the 95% confidence level. The Tukey test was used to assess the statistical significance of these differences. It should be noted that the same method was used to compare the statistical significance of differences between the data in the other figures presented later in the article." **(Lines 1053–1060, page 38)**

"The average concentrations of most FAs were significantly ($p < 0.05$) higher in autumn and winter than in spring and summer (Figure 2a), indicating a substantial increase in fatty acid emissions during autumn and winter. The LFA/HFA and C18:0/C16:0 ratios were lower in autumn and winter than in summer (Figures S3b and S3c)." **(Lines 222–225, page 9)**

Similar modifications and annotations can be found in Figures 3, 4, S3, and S4, as well as in the related discussions of seasonal differences throughout the manuscript.

**Specific comments 3.** Discussion: The discussion would benefit from expanded comparisons with similar studies conducted in other Chinese cities. Additionally, a more in-depth exploration of the mechanisms behind seasonal variations (instead of merely presenting the results of compounds one by one) should be included.

**Response:** Thank you for your constructive suggestions. Following your advice, we have made two key improvements:

1. Expanded comparisons with other chinese cities:

We have broadened the discussion to compare our findings with results from studies in other urban areas in China, as well as relevant coastal, marine, and polar environments. For example:

Fatty acids (FAs): "The total FA concentration was in a range of 3.24–657.86 ng m$^{-3}$, with an average concentration of 196.50 ± 110.92 ng m$^{-3}$. Similar molecular distribution patterns and concentrations have been documented in urban aerosol studies across China (59–2,090 ng m$^{-3}$; He et al., 2004 and 2006; Wang et al., 2006; Haque et al., 2019; Fan et al., 2020). In these Chinese cities, FAs are primarily influenced by biogenic sources, including plant releases and biomass burning, as well as fossil fuel combustion and residential cooking. This average concentration generally exceeds those observed for coastal and marine aerosol samples (0.1–160 ng m$^{-3}$; Kawamura et al., 2003; Wang et al., 2007; Fu et al., 2011), which are primarily influenced by marine biological activities and long-range transport of continental sources." **(Lines 191–198, page 8)**

Fatty alcohols: "The total concentration of n-alkanols was in a range of 5.80–572.01 ng m$^{-3}$ (average = 113.99 ± 92.50 ng m$^{-3}$). This concentration fell within the range of reported n-alkanol concentrations in urban aerosols in China (3.1–1,301.0 ng m$^{-3}$; Wang et al., 2006; Haque et al., 2019; Fan et al., 2020) and generally exceeded that of coastal and marine aerosol samples (0.1– 19.7 ng m$^{-3}$; Kawamura et al., 2003; Fu et al., 2011). In urban aerosols of China, fatty alcohols are primarily derived from vegetation releases, biomass burning, and resuspension of soil particles. The fatty alcohols in coastal and marine aerosols are primarily attributed to the long-range transport of terrestrial soil and biomass burning as well as marine biogenic sources." **(Lines 237– 244, pages 9–10)**

Biomass burning tracers (e.g., levoglucosan): "Levoglucosan—a prominent biomarker for biomass burning—has been extensively investigated in urban aerosol samples, with concentrations in a range of 22–2,706 ng m$^{-3}$ (Wang and Kawamura, 2005; Wang et al., 2006; Fu et al., 2010; Haque et al., 2019; Fan et al., 2020), indicating the significant influence of biomass burning on the urban atmosphere. The widespread detection of levoglucosan in various environments—including suburban (10–482 ng m$^{-3}$; Yttri et al., 2007; Fu et al., 2008), marine (0.2–30 ng m$^{-3}$; Simoneit et al., 2004a; Fu et al., 2011), and polar regions (0–3 ng m$^{-3}$; Stohl et al., 2007; Fu et al., 2009)—indicates its potential for long-range atmospheric transport. Compared to those of the major polar organic compounds found in marine environments, such as FAs, fatty alcohols, and saccharides, the concentrations of these substances in urban PM$_{2.5}$ are substantially higher. This is primarily attributed to the substantial impact of anthropogenic activities in urban

environments—such as vehicle emissions, industrial processes, and residential heating and cooking—all of which can release these major polar components (Wang et al., 2006; Fu et al., 2008; Haque et al., 2019; Fan et al., 2020). In contrast, the organic matter in marine $PM_{2.5}$ originates from relatively simpler sources with lower emissions, mainly influenced by natural processes, including marine biological activity and the long-range transport of continental pollutants (Kawamura et al., 2003; Simoneit et al., 2004a; Wang et al., 2007; Fu et al., 2011)." **(Lines 262–276, pages 10–11)**

SOC marker ratios: "This relatively high C5/MTL, 2-MGA/MTL, or MTL2/MTL ratio observed during the cold season compared to that in the warm season is commonly reported in urban environments across China, including Beijing (Liu et al., 2023), Tianjin (Wang et al., 2019; Fan et al., 2019), Nanjing (Yang et al., 2022), Shanghai (Yang et al., 2022), and Guangzhou (Yuan et al., 2018). This phenomenon is primarily attributed to higher $NO_x$ concentrations and increased aerosol acidity in the cold season versus the warm season in Chinese cities, which promote the formation of secondary C5-alkene triols, 2-MGA, and MTL2." **(Lines 423–429, pages 16–17)**

2. Expanded discussion of seasonal variation mechanisms:

In addition to comparing seasonal differences, we have analyzed and discussed in detail the mechanisms driving observed seasonal patterns. For example:

Fatty acids (FAs): "The average concentrations of most FAs were significantly ($p < 0.05$) higher in autumn and winter than in spring and summer (Figure 2a), indicating a substantial increase in fatty acid emissions during autumn and winter. The LFA/HFA and C18:0/C16:0 ratios were lower in autumn and winter than in summer (Figures S3b and S3c). This indicated that in autumn and winter, FAs were significantly influenced by terrestrial plant or biomass burning, whereas in summer, they were more influenced by vehicular emissions, marine phytoplankton, and cooking emissions. Conversely, the (C16:1 + C18:1)/(C16:0 + C18:0) values were higher in winter than in summer (Figure S3d). This suggested that unsaturated FAs experienced a greater degree of photochemical degradation during the warm seasons. Unsaturated FAs can be rapidly oxidized by ozone or OH radicals. The higher temperatures, relatively stronger solar radiation, and higher concentrations of $O_3$ and OH radicals in the summer season will facilitate the oxidative decomposition of unsaturated FAs (Wang et al., 2006; Fan et al., 2020)." **(Lines 222–232, page 9)**

Fatty alcohols: "The average concentrations of most fatty alcohols were significantly ($p < 0.05$) higher in autumn and winter than in summer (Figure 2b), indicating a substantial increase in fatty alcohols emissions during autumn and winter. The LMW/HMW values were relatively low in autumn and winter and higher in spring and summer (Figure S3f). This suggests an increased contribution of soil and marine organisms to n-alkanols in spring and summer. Contributions from plant or biomass burning increased in autumn and winter. In Chinese cities, rising temperatures in spring increase soil microbial activity, with marine air masses exerting a stronger influence during summer, potentially increasing emissions from soil and marine organisms in these seasons. In contrast, autumn and winter coincide with the crop harvest and increased winter heating, leading to likely increases in biomass burning emissions during these periods (Wang et al., 2006; Haque et al., 2019; Fan et al., 2020)." **(Lines 249–257, page 10)**

Biomass burning tracers: "The Lev/OC and Lev% values were higher in autumn and winter than in spring and summer (Figures S3i and S3j), further indicating that biomass burning contributions increased during the cold months. As previously noted, the primary reason for increased biomass burning emissions during the cold season is that autumn marks the crop harvest in China, leading to more straw burning. Furthermore, the temperature decrease in winter results in additional biomass burning for heating purposes." **(Lines 318–323, page 12–13)**

SOC markers: "In summer, the concentrations of the hydroxy acids exceeded those in other seasons (Figure 3b). A positive correlation was observed between the polyacids and temperature (r = 0.35–0.51, p < 0.01), suggesting enhanced secondary photochemical oxidation in the warm months. The secondary formation of hydroxy acids in the atmosphere is suggested to be related to the ozone-driven oxidation of organic compounds. Elevated ozone levels and stronger solar radiation during the summer season create more favorable conditions for the secondary production of hydroxy acids (Kawamura and Ikushima, 1993; Kawamura and Sakaguchi, 1999; Claeys et al., 2004)." **(Lines 362–368, page 14)**

Phthalic acid: "The phthalic acid concentrations were higher in autumn and winter, whereas terephthalic acid levels peaked in spring and summer (Figure 3b). This indicates that the secondary transformation of PAHs was more pronounced during the cold months, whereas high temperatures in spring and summer promoted the volatilization and transformation of phthalates from plastic sources. Phthalates are employed as plasticizers in resins and polymers. Because they are not chemically bound to the polymer matrix, they can be easily released into the air through evaporation. Higher ambient temperatures in spring and summer favor phthalate release from the matrix (Wang et al., 2006; Fu et al., 2010)." **(Lines 383–389, page 15)**

SOC marker ratios "The MBTCA/3-HGA value was higher in spring (1.31 $\pm$ 0.66) and summer (1.45 $\pm$ 0.70) than in autumn (1.09 $\pm$ 0.72) and winter (0.88 $\pm$ 0.32) (Figure S5e). This suggests that the contribution of α-pinene to monoterpene was higher in spring and summer than in autumn and winter. This phenomenon can be attributed to the association of 3-HGA formation with $NO_x$-rich environments, where the relatively high $NO_x$ concentrations in autumn and winter (Figure S2) may enhance 3-HGA production. The ratio of the total isoprene to monoterpene oxidation products (Iso/Pine) was lower in autumn and winter than in summer (Figure S5f). This pattern is also commonly observed in other urban environments of China, suggesting that the higher temperatures in summer favored isoprene release (Li et al., 2018; Fan et al., 2020; Liu et al., 2023)." **(Lines 453–460, page 17–18)**

Together, these mechanistic insights highlight how emission sources, atmospheric processing, and seasonal meteorological conditions drive the observed variation in organic aerosol composition.

**Specific comments 4.** Number of Figures: The manuscript contains as many as ten figures, which is excessive. It is recommended to streamline the number to around five. For instance, Figure 1a does not present original results and mainly supports the introduction; it should be moved to the supporting information. Figure 1b contributes minimal information and could be removed entirely. A similar review should be applied to the remaining figures to ensure only important Figures with enough essential data are included in the main text.

**Response:** Thank you very much for your comments. In response, we have revised Figure 1: Figure 1a has been moved to the Supplementary Materials to better support the introduction, and Figure 1b has been removed, as it contributed minimal information. Following a thorough review, we have reduced the number of figures in the main text from 10 to 7. Several figures that were previously in the main text have now been relocated to the Supplementary Materials, ensuring that only the most essential data and results remain in the main text.

**Specific comments 5.** Language and Terminology: The manuscript would be more accessible if complex sentences were simplified and terminology made consistent. For example, the sentence: "The results indicated that the concentrations of fatty acids, fatty alcohols, and saccharides were relatively high, whereas lignin and resin products, sterols, glycerol, hydroxy acids, and aromatic acids were present at low concentrations" (lines 16–18) could be simplified to: "Fatty acids, fatty alcohols, and saccharides showed relatively high concentrations, while lignin, resin products, sterols, glycerol, hydroxy acids, and aromatic acids were low."

**Response:** Thank you for your valuable suggestion regarding language and terminology. We agree that simplifying complex sentences and ensuring consistent terminology will improve clarity and accessibility. As you suggested, we have revised the example sentence as follows:

Original: "The results indicated that the concentrations of fatty acids, fatty alcohols, and saccharides were relatively high, whereas lignin and resin products, sterols, glycerol, hydroxy acids, and aromatic acids were present at low concentrations."

Revised: "Fatty acids (FAs), fatty alcohols, and saccharides showed higher concentrations than lignin, resin products, sterols, glycerol, hydroxy acids, and aromatic acids;" **(Lines 17–18, page 1)**

We also have carefully reviewed the entire manuscript and made similar simplifications and terminology adjustments throughout to enhance readability and consistency.

**Specific comments 6.** Statistical Analysis: The manuscript frequently compares data across seasons and sources (e.g., LMW/HMW ratios, mannosan and galactosan concentrations, L/M and M/G ratios in Figures 3–7). However, there is no indication of statistical significance in the text or figures. The authors should include appropriate statistical tests to support these comparisons.

**Response:** Thank you for your insightful comment regarding statistical analysis. In response, we have revised Figures 3–7 (now renumbered as Figures 2–4, S3, and S5 in the revised manuscript) to incorporate statistical significance indicators. Specifically, we now use different letters above the bars to denote significant differences in seasonal or source-related data, as determined by the Tukey test at a 95% confidence level. Similar modifications and annotations can be found in the related discussions of seasonal differences throughout the manuscript.

**Specific comments 7.** Line 91: Please provide the sampling flow rate used during the study.

**Response:** Thank you for pointing this out. We have added the sampling flow rate information to the manuscript. The revised sentence now reads:

"An 8-inch $\times$ 10-inch quartz fiber filter (Pall Tissuquartz, USA) was used in a high-volume air sampler at a flow rate of 1.05 $\pm$ 0.03 m$^3$ min$^{-1}$ for sample collection" (**Lines 91–92, page 4).**

**The authors would like to express their sincere gratitude to the editor and reviewers for their valuable suggestions and comments, which have greatly improved the quality of this manuscript.**

---

## Referee Report (RR1)

Summary: This manuscript describes measurements of polar organic species in Nanchange, China and source apportionment of organic carbon (OC) using tracer based and chemical mass balance (CMB) approaches. The approaches taken by the authors are relatively standard, and the integration of multiple approaches strengthens the paper. The paper reports insights to annual variations in primary and secondary sources. Further insight is provided into the characteristics of OC during winter pollution episodes, when coal burning and secondary aerosol had relatively larger impacts. The figures are very detailed and contain a lot of information. There are numerous aspects of the manuscript, detailed below, that should be addressed prior to publication.

Overall comments that should be addressed throughout the manuscript:

- 1. In applying the "tracer-based" method to source apportionment (page 5), the results are highly dependent on the source profiles utilized. It is best practice to use locally-sourced profiles, when available, and those that are representative of the relevant sources at the time of the study. There is no justification provided for the selection of profiles beyond that they were utilized by Kleindienst et al. in their 2007 and 2012 publications. The authors must provide justification for the selected profiles, discuss their representativeness, and the potential errors introduced by these selections. Additionally, the authors should make a diligent effort to utilize the most relevant, regionally-specific, and up-to-date information when available, noting that the profiles used in these studies can be 20 years old. For example, regional profiles for straw burning should be used, given the conclusions of the authors of the importance of this source (10.1016/s1001-0742(07)60027-8). Regional profiles for other relevant sources are available and should be considered for robust results.
- 2. In applying the CMB approach to source apportionment, there is likewise a need for discussion of the selected profiles, their representativeness, and uncertainties introduced by differences in these profiles and local and/or regional sources.
- 3. The authors should also specifically state which chemical species were used in the CMB model and provide justification for these choices and discussion of what sources are and are not represented. The extraordinarily good agreement between the tracer-based and CMB results implies that only a few fitting species may have been used, which means that the CMB model may not be well-constrained.
- 4. Improve readability to integrate results from multiple tracers, rather than treating them one by one (i.e. pages X to Y). For example, integrating results from
- 5. There is a sizable portion of OC that is not attributed to the primary and secondary sources considered. This requires further discussion is it due to a mismatch of the selected tracer-to-OC fractions (or source profiles) to the ambient data? Are

- important sources in the region not included or considered? If a major secondary organic aerosol source has not been considered, then statements regarding the dominance of primary over secondary sources are not accurate.
- 6. The authors use linear correlations as a tool for data analysis, which assumes that date are normally distributed. Are the data, in fact, normally distributed? Please perform a statistical test for normality and include that result in the discussion.
- 7. In reporting organic species concentrations (i.e. lines 191, 192, and elsewhere), consider the appropriate number of significant figures. Likely 2-3 digits are statistically significant (considering uncertainties in the range of 10%) and should be reported, rather than 5.
- 8. Application of the "tracer-based" method and CMB approaches to source apportionment assume conservation of mass between source and receptor. However, the fatty acid results indicate that "unsaturated FAs underwent significant photochemical degradation" (line 220). Significant chemical transformations would lead to errors in source attribution. The influence of chemical transformations on source apportionment must be discussed.
- 9. A more thorough discussion of the limitations of the current work are needed.

**Specific comments:**

- 10. It would be helpful if the authors could clarify in the abstract (lines 15-16) their approach to "comprehensive analysis" of polar compounds and source attribution to primary/secondary and anthropogenic/natural sources.
- 11. At lines 92-94, indicate the specific locations of the meteorological and gas sensors and their relation to the PM sampling site.
- 12. Lines 99-100, include a reference to the IMPROVE protocol used for EC and OC analysis.
- 13. Line 106, justification is needed for the use of n-alkanes as internal standards for polar compounds, especially because the alkanes do not undergo silylation derivitazation.
- 14. The statements about OC/EC values from 172-176 do not seem to consider secondary sources, or mixtures of sources. This seems contradictory with the other results in the study and should be removed.
- 15. Line 268, the authors should also consider reports of chemical degradation of levoglucosan in the atmosphere, for example, and how this may influence their source apportionment results.
- 16. In section 3.4.1, the authors report concentrations of C5 alkene triols. The authors should consider more up to date information available in the literature regarding this group of compounds (i.e. Frauenheim, et al. doi/10.1021/acs.estlett.2c00548).

- The majority of these "triols" have been demonstrated to be artifacts, and a structure with a ring rather than a double bond is major isomer.
- 17. Figure 5, S6, F7. The text in the legends is very small and difficult to read. The important information that distinguishes the various sources is sub-scripted and difficult to see. Please enlarge the text in the legend to improve readability.
- 18. Lines 527-528, please explain how OC was converted into OA.

---

## Author Response (AR2)

**Response to the reviewers' comments point-by-point**

**Reviewer 3:**

General comments. This manuscript describes measurements of polar organic species in Nanchang, China and source apportionment of organic carbon (OC) using tracer based and chemical mass balance (CMB) approaches. The approaches taken by the authors are relatively standard, and the integration of multiple approaches strengthens the paper. The paper reports insights to annual variations in primary and secondary sources. Further insight is provided into the characteristics of OC during winter pollution episodes, when coal burning and secondary aerosol had relatively larger impacts. The figures are very detailed and contain a lot of information. There are numerous aspects of the manuscript, detailed below, that should be addressed prior to publication.

Overall comments that should be addressed throughout the manuscript:

Overall comments 1. In applying the "tracer-based" method to source apportionment (page 5), the results are highly dependent on the source profiles utilized. It is best practice to use locally-sourced profiles, when available, and those that are representative of the relevant sources at the time of the study. There is no justification provided for the selection of profiles beyond that they were utilized by Kleindienst et al. in their 2007 and 2012 publications. The authors must provide justification for the selected profiles, discuss their representativeness, and the potential errors introduced by these selections. Additionally, the authors should make a diligent effort to utilize the most relevant, regionally-specific, and up-to-date information when available, noting that the profiles used in these studies can be 20 years old. For example, regional profiles for straw burning should be used, given the conclusions of the authors of the importance of this source (10.1016/s1001-0742(07)60027-8). Regional profiles for other relevant sources are available and should be considered for robust results.

**Response:** We thank the reviewer for the careful and constructive comment on the selection and representativeness of source profiles and tracers. In response, we have made targeted revisions. Both source-apportionment methods applied in this study—the CMB model and the tracer-based approach—depend critically on the choice and interpretation of source profiles and representative chemical species. Recognizing that the original manuscript provided limited detail on this point, we have substantially expanded and clarified the relevant material in the revised manuscript

First, we expanded Section 2.3 (Source apportionment methods) to provide a detailed description of the source profiles and representative tracers used in both the CBM model and tracer-based approaches. Where available, we prioritized regional and local profiles that best match the study area and observation period. Specific revisions are as follows:

"To quantify the contributions of various primary sources to OC and PM2.5, i.e., the POC and POA, we utilized the CMB model (version 8.2), a widely accepted source apportionment method developed by the United States Environmental Protection Agency (Lewandowski et al., 2008;

Stone et al., 2009; Guo et al., 2012; Wu et al., 2020; Xu et al., 2021). The model assumes that the chemical composition of pollutants remains unchanged during transport, allowing measured chemical species at the receptor to be expressed as the linear sum of contributions from individual source categories. Accurate CMB results depend critically on the selection of representative emission sources and chemical tracers. These must include all major contributors and feature chemical markers that are stable during transport and distinct among sources (Stone et al., 2009). To meet these requirements, we first identified four major primary OC sources: biomass burning, coal combustion, vehicular exhaust, and cooking emissions, based on China's atmospheric PM and gaseous pollutant emission inventories (Huang et al., 2015; Li et al., 2019; Tong et al., 2020), We also considered the contributions of plant debris and fungal spores to OC, base on the regional studies on OC sources (Fan et al., 2020; Xu et al., 2020; Wu et al., 2020). Representative source profiles and tracers of these sources were compiled from recent local and regional reports. For example, biomass burning releases substantial amounts of levoglucosan, accounting for ~8.3% of OC in local profiles (Zhang et al., 2007), and was therefore chosen as its marker compound. For coal combustion and vehicular exhaust, both characterized by complex mixtures include n-alkanes, polycyclic aromatic hydrocarbons (PAHs), and hopanes (Rogge et al., 1993a; Oros and Simoneit, 2000; Zhang et al., 2008; Cai et al, 2017). Local emission profiles indicated coal combustion tends to emit a higher proportion of 3- to 4-ring PAHs (Zhang et al., 2008), whereas vehicular exhaust is characterized by higher contributions from C20-C22 n-alkanes (Cai et al, 2017). Consequently, we selected 3- to 4-ring PAHs and C20-C22 n-alkanes as the representative organic markers for local coal combustion and vehicle emissions, respectively. Cooking emissions are dominated by fatty acids, particularly saturated palmitic acid (C16:0) and stearic acid (C18:0), as well as unsaturated fatty acids like palmitoleic acid (C16:1) and oleic acid (C18:1). Given the instability of unsaturated fatty acids in ambient air (Kawamura and Gagosian, 1987; Rudich et al., 2007), only the more stable saturated fatty acids—palmitic and stearic acids—were used as characteristic markers for local cooking emissions (He et al., 2004; Zhao et al., 2007 and 2015). For plant debris, emission profiles from Los Angeles area indicate that plants release considerable quantities of long-chain odd-carbon-number n-alkanes (e.g., C25, C27, C29) (Rogge et al., 1993b); however, many local studies indicate that biomass burning, coal combustion, and vehicle exhaust also emit long-chain n-alkanes to some extent (Zhang et al., 2007, 2008; Cai et al., 2017). Overlap among sources diminishes the diagnostic value of long-chain n-alkanes, some investigations have consequently proposed glucose as a more selective tracer for plant debris (Fan et al., 2020; Xu et al., 2021). Puxbaum and Tenze-Kunit (2003) report that glucose comprises approximately 5.2% of plant-derived organic carbon, whereas other sources other than plants rarely emit glucose. On this basis, glucose was adopted here as the plant emission biomarker. Additionally, certain fungal spores contribute to OC in ambient air; literature reports indicate that particles originating from fungal spores contain abundant mannitol and arabitol, accounting for roughly 13% and 19% of OC, respectively (Bauer et al., 2002, 2008). Accordingly, mannitol and arabitol were chosen as the marker compounds representing fungal spores in this study. Detailed source profile information used in CMB model is provided in Fig. S3 of the supplementary material. It should be noted that the local source profile dataset subdivides major sources—such as coal combustion, vehicle exhaust, and cooking-into multiple categories (for example, industrial versus residential coal burning, gasoline versus diesel exhaust, and regional cooking styles such as Guangdong versus Sichuan). For each major source category, we represented typical source characteristics by using

either the mean profile of its subcategories or the average profile derived from multiple observational studies. Similarly, using the proportions of characteristic species in  $PM_{2.5}$  from these emission source profiles, the CMB model subsequently estimated each source's contribution to  $PM_{2.5}$ . In the primary emission profiles applied here, the PM-to-OC ratio ranged from 1.42 to 2.15." (Lines 126–172, pages 6–7).

"To assess the contributions of various secondary sources to OC and  $PM_{2.5}$ , i.e., SOC and SOA concentrations, we employed tracer-based method, a well-established method for SOC and SOA source apportionment. The tracer-based method operates on principles analogous to those involved in the CMB model, relying on the mass fractions of characteristic tracers in SOC ( $f_{SOC}$ ) and SOAs ( $f_{SOA}$ ) from emission sources to ascertain the contributions of different sources. The SOC and SOA tracer-based method was first proposed and utilized by Kleindienst et al. (2007 and 2012). Its specific calculation procedure is as follows:

$$[SOC] = \frac{\sum_{i} [tr_i]}{f_{SOC}},$$
(1)

$$[SOA] = \frac{\sum_{i} [tr_i]}{f_{SOA}},\tag{2}$$

where  $\sum$ i[tri] is the total concentration of the selected tracers in the sample, denoting representative compounds from specific emission categories;  $f_{SOC}$  and  $f_{SOA}$  are the mass fractions of the tracers in OC and PM2.5 from secondary emissions, respectively. The values of  $f_{SOC}$  and  $f_{SOA}$  are determined according to the chamber experiments conducted, in which gaseous precursors are transformed into SOC and SOAs under oxidative and illuminative conditions. Herein, we employed six compounds associated with hemiterpenes, four compounds related to monoterpenes, as well as  $\beta$ -caryophyllinic acid, 2,3-dihydroxy-4-oxopentanoic acid, and phthalic acid as marker compounds to indicate the contributions of different biogenic (hemiterpenes, monoterpenes, and sesquiterpenes) and anthropogenic (toluene and naphthalene) sources to SOC and SOA. We adopted the  $f_{SOC}$  and  $f_{SOA}$  values used by Kleindienst et al. (2007 and 2012). to calculate the SOC and SOA concentrations originating from various sources. All marker selections, f-values, and the conversion coefficients between  $f_{SOA}$  and  $f_{SOC}$  derived from chamber studies are listed in Table S2, and the calculations assume that the chamber-derived SOC and SOA relationships are representative of the relevant atmospheric oxidation regimes, acknowledging associated uncertainties." (Lines 173–193, pages 7–8)

Second, we have added a detailed explanation and discussion of the errors and uncertainties introduced by the choice of source profiles and representative tracers in Section 3.5 (Source apportionment of OC and aerosol). The specific content is as follows:

"The calculation of POC and SOC contributions based on the CMB model and tracer-based method inherently involves uncertainties. For the CMB model, a primary source of uncertainty is the variability of tracer mass fractions within OC ( $f_{OC}$ ) from the same primary emission sources across various observational studies. For example, reported levoglucosan/OC fractions range from 12% (Andreae, et al., 2001, 2019) to 13% (Zheng et al., 2002) and 16% (Fine et al., 2004), whereas the local source profile we reference suggests 8.3% (Zhang et al., 2007). Since recent, site-specific emission data are often unavailable, the choice of source profile and corresponding  $f_{OC}$  values can strongly influence apportionment outcomes. Secondly, a fundamental assumption

of the CMB model is that the selected markers representing various sources remain stable during atmospheric transport and do not undergo significant chemical transformation. However, truly conservative species are rare. For example, levoglucosan, commonly used as a biomass-burning marker and generally considered relatively stable (Wan et al., 2021), has nonetheless been shown to decrease during long-range transport through wet and dry deposition (Fu et al., 2011) and by photodegradation (Holmes and Petrucci, 2006; Stohl et al., 2007; Hoffmann et al., 2010). Analogous loss or alteration of other source tracers can bias CMB results, tending to underestimate contributions from sources whose markers are degraded or removed during transport. For tracer-based methods, the appropriateness of the selected fSOC and fSOA values is also crucial to the accuracy of the results. Compared with primary source profiles, direct observational information on secondary source profiles is sparse. This paucity reflects the experimental difficulty of reproducing atmospheric oxidation and photochemical ageing under representative light and precursor conditions in laboratory or chamber systems. Most studies employing tracer-based methods to estimate SOC and SOA contributions have relied on fSOC and fSOA values reported by Kleindienst et al. (2007, 2012), which are now relatively dated. SOC and SOA formation are influenced not only by oxidants and sunlight but also by factors such as relative humidity, precursor concentrations, NOx levels, and other ambient variables. The chamber conditions cannot fully reproduce the complexity of real atmospheres, leading to inevitable discrepancies in fSOC and fSOA values between chamber-derived results and ambient air (Fu et al., 2009; Guo et al., 2012; Ding et al., 2012 and 2014; Haque et al., 2023). Additionally, the  $f_{SOC}$  and fSOA values used were average, and their inherent standard deviations could result in deviations of 21% to 48% in the calculated SOC contributions from different sources (Kleindienst et al. 2007 and 2012). Despite the inherent uncertainties associated with both the CMB model and tracer-based method, these approaches remain convenient tools for estimating the contributions of various POC and SOC, yielding relatively reasonable results (Stone et al., 2009; Ding et al., 2012 and 2014; Al-Naiema et al., 2017; Ren et al., 2021; Xu et al., 2021; Haque et al., 2023). However, further efforts are needed to reduce these uncertainties. One critical step will be to conduct more extensive, site-specific observational studies of different types of primary and secondary emission sources. Such data are essential for identifying representative tracers within different sources and accurately determining their mass fractions in OC and OAs. Additionally, when fSOC and fSOA values are applied in both CMB model and tracer-based calculations, using the most recent, locally relevant emission source profile to minimize uncertainties is advisable." (Lines 599–634, pages 23-24)

**Overall comments 2.** In applying the CMB approach to source apportionment, there is likewise a need for discussion of the selected profiles, their representativeness, and uncertainties introduced by differences in these profiles and local and/or regional sources.

**Response:** We thank the reviewer for this important comment. As noted in our response to Comment 1, we have expanded Section 2.3 (Source apportionment methods) to describe in detail the source profiles used in the CMB analysis, the rationale for their selection, and the representative tracers chosen. We have also expanded Section 3.5 (Source apportionment of OC and Aerosol) to provide a thorough discussion of the uncertainties and potential errors introduced by differences in source profiles and tracer representativeness. Please refer to our response to Comment 1 and the revised manuscript (Sections 2.3 and 3.5) for the full details.

**Overall comments 3.** The authors should also specifically state which chemical species were used in the CMB model and provide justification for these choices and discussion of what sources are and are not represented. The extraordinarily good agreement between the tracer-based and CMB results implies that only a few fitting species may have been used, which means that the CMB model may not be well-constrained.

**Response:** Thank you for this comment. In the original manuscript we estimated POC and SOC using the tracer-based method and compared those estimates with CMB results. In the revised manuscript we treat the two approaches independently: the CMB is used to apportion POC sources, while the tracer-based method is applied separately to estimate SOC fractions. Because the methods are now applied to different OC fractions, the previous mutual-validation comparisons between the tracer-based method and the CMB have been removed. As noted in our responses to the comment 1 and 2, we have expanded Section 2.3, "Source apportionment methods," to provide a detailed description of the source profiles employed in the CMB analysis and the representative chemical species selected, together with justification for these choices.

**Overall comments 4.** Improve readability to integrate results from multiple tracers, rather than treating them one by one (i.e. pages X to Y).

**Response:** Thank you for this valuable suggestion. We have reorganized the Results to present tracer information in a more integrated and synthetic way. Specifically, primary-emission tracers are discussed in grouped categories (e.g., sugar compounds are considered together; lignin, resin-derived products and sterols are discussed as a unit; glycerol, hydroxy acids and aromatic acids are treated collectively). Secondary-emission tracers are likewise grouped into anthropogenic and biogenic classes and interpreted together to highlight coherent source and formation patterns. These changes improve readability and emphasize cross-tracer consistency; please see the revised manuscript Sections 3.2-3.4 (Major polar components, Minor polar components, and SOA tracers in  $PM_{2.5}$ ) for the updated presentation.

**Overall comments 5.** There is a sizable portion of OC that is not attributed to the primary and secondary sources considered. This requires further discussion—is it due to a mismatch of theselected tracer-to-OC fractions (or source profiles) to the ambient data? Are important sources in the region not included or considered? If a major secondary organic aerosol source has not been considered, then statements regarding the dominance of primary over secondary sources are not accurate.

**Response:** We thank the reviewer for highlighting the portion of OC (other OC) that remains unattributed. We have added detail analysis and discussion of other OC in the revised manuscript. The relevant discussion is expressed in the updated manuscript as follows:

"The fraction of OC that remains unexplained by the CMB model and tracer-based methods and is therefore classified as "other OC". This component is defined as the difference between measured OC and the summed contributions of POC and SOC from all apportioned sources estimated by the CMB model and tracer approaches. On average, other OC accounts for ~34% of measured OC (Figures. 5c, 5f). The presence of other OC likely reflects limitations in source identification:

some OC forms (e.g., liquid-phase OC or aged primary OC) are difficult to capture with either the CMB model or tracer techniques (Ding et al., 2014; Fan et al., 2019; Wu et al., 2020; Xu et al., 2021). Although absolute concentrations of other OC are lower in spring and summer than in autumn and winter, its fractional contribution shows the opposite seasonal pattern—about 45%–50% in spring and summer versus 12%-29% in autumn and winter (Figures. 5c, 5f). Similar patterns have been reported elsewhere in other citie of China, e.g., Beijing, where estimated concentrations of other OC are higher in winter but its proportion is greater in summer (~44% vs. 22-25% in winter) (Wang et al., 2009; Wu et al., 2020; Xu et al., 2021). Previous studies indicate that this "other OC" is likely dominated by SOC (Guo et al., 2012; Wu et al., 2020; Xu et al., 2021). To assess whether other OC is predominantly primary or secondary, we used an EC-based method to estimate total POC and SOC; unlike tracer methods, the EC approach simply partitions OC into POC and SOC and therefore can include unapportioned components. Accordingly, the POC calculated via EC-based method minus the POC calculated via CMB model yields the "other POC" (unapportioned POC), and similarly, SOC calculated via EC-based method minus the tracer-based SOC yields the "other SOC" (Figure. S8). Our results show that, across all four seasons, the concentration of other SOC exceeds that of other POC (Figures. S8a, S8b), indicating that these unidentified other OC components are likely predominantly SOC. The concentration of other SOC is substantially higher in autumn and winter—especially during pollution periods in winter—highlighting the critical role of SOC in aerosol pollution during these seasons (Figures. S8c, S8d). Conversely, the proportion of other SOC during summer is the highest among all seasons (around 32%) (Figure. S8b), suggesting that elevated temperatures and intense radiation during summer enhance SOC formation efficiency (Wu et al., 2020; Xu et al., 2021). Although the other SOC fraction exceeds other POC, EC-based estimates of total SOC remain smaller than total POC in all seasons. Similarly, SOC estimated by the tracer method is markedly lower than POC apportioned by the CMB model. These results together indicate that primary emissions play a significant role in urban OC pollution, with substantial POC contributions forming the basis for elevated OC concentrations. However, the influence of SOC should not be overlooked. Numerous studies have shown that even minor increases in SOC during winter pollution episodes can exacerbate OA pollution (Li et al., 2019; Xu et al., 2021; Haque et al., 2023)." (Lines 560-590, pages 21-22)

**Overall comments 6.** The authors use linear correlations as a tool for data analysis, which assumes that date are normally distributed. Are the data, in fact, normally distributed? Please perform a statistical test for normality and include that result in the discussion.

**Response:** Thank you very much for your valuable comment. In the revised manuscript, all data used for linear fitting were first tested for normality. Specifically, the Shapiro-Wilk test was applied to assess the normality of the data. Relevant explanations and discussions can be found in Section 3.6 (Characteristics of OAs during winter pollution) (**Lines 662–664, page 25**). Additionally, figures that include linear fits (Figures 7 and S12) now include the following note: "Before conducting linear correlation analyses, the Shapiro-Wilk test was used to assess the normality of the data, and linear correlations were conducted only for datasets significantly conforming to a normal distribution (p > 0.05)."

Overall comments 7. In reporting organic species concentrations (i.e. lines 191, 192, and

elsewhere), consider the appropriate number of significant figures. Likely 2-3 digits are statistically significant (considering uncertainties in the range of 10%) and should be reported, rather than 5.

**Response:** Thank you for this helpful suggestion. In the revised manuscript, concentration values are reported with two to three significant figures, consistent with the estimated uncertainties. This unified format has been applied consistently throughout the text, tables, figures, and Supplementary Information.

**Overall comments 8.** Application of the "tracer-based" method and CMB approaches to source apportionment assume conservation of mass between source and receptor. However, the fatty acid results indicate that "unsaturated FAs underwent significant photochemical degradation" (line 220). Significant chemical transformations would lead to errors in source attribution. The influence of chemical transformations on source apportionment must be discussed.

**Response:** Thank you very much for your valuable comment. Considering that unsaturated fatty acids are prone to degradation during the transfer process, we excluded unsaturated fatty acids when selecting fatty acids as characteristic compounds for cooking emission sources. The relevant revisions are as follows:

"Cooking emissions are dominated by fatty acids, particularly saturated palmitic acid (C16:0) and stearic acid (C18:0), as well as unsaturated fatty acids like palmitoleic acid (C16:1) and oleic acid (C18:1). Given the instability of unsaturated fatty acids in ambient air (Kawamura and Gagosian, 1987; Rudich et al., 2007), only the more stable saturated fatty acids—palmitic and stearic acids—were used as characteristic markers for local cooking emissions (He et al., 2004; Zhao et al., 2007 and 2015)." (lines 148–152, page 6)

**Overall comments 9.** A more thorough discussion of the limitations of the current work are needed.

**Response:** Thank you very much for this important suggestion. We have expanded the Conclusion to include an in-depth consideration of the study's limitations (please see Conclusion, new paragraph). The added text is given below:

"The study findings underscore the necessity for targeted management strategies that consider primary and secondary anthropogenic emission sources across different seasons and pollution periods. Although the CMB model and tracer-based method provided preliminary insights into the OC and OA from diverse sources, the study encountered several inherent limitations. First, the primary emission source profiles employed in the CMB model exhibit variability across different studies and regions, making it challenging to establish standardized source characteristic parameters and potentially affecting the accuracy of source apportionment. Second, the proportions of SOC tracers obtained from laboratory chamber experiments are influenced by various factors, and incorporating these proportions into tracer-based methods may introduce potential biases or uncertainties in the estimates. To address these issues, future research should prioritize comprehensive observational studies of primary emission sources to obtain

high-resolution, region-specific emission data, thereby improving the applicability of source profiles. Combining field observations with laboratory simulations can also provide a more accurate characterization of secondary emissions, ultimately reducing the uncertainties associated with tracer-based estimates of SOC contributions." (lines 694–706, pages 26–27)

**Specific comments 10.** It would be helpful if the authors could clarify in the abstract (lines 15-16) their approach to "comprehensive analysis" of polar compounds and source attribution to primary/secondary and anthropogenic/natural sources.

**Response:** Thank you for these valuable comments. We have added a description of the relevant analytical methods in the abstract. Specific biomarkers and diagnostic ratios were applied to characterize OA sources and distribution patterns, while chemical mass balance (CMB) models and tracer-based approaches were used to estimate source contributions. Statistical analyses were conducted to investigate OA characteristics and drivers during winter pollution episodes. The revised abstract is as follows:

"Due to the complex composition of organic aerosols (OAs), identifying their sources and understanding their dynamics remain challenging, particularly in urban environments of China where natural and anthropogenic influences to OAs intersect. This study aimed to clarify the relative contributions of primary emissions and secondary formation to urban OAs and confirm the sources and influencing factors of OA pollution. We analyzed major polar organic compounds in fine particulate matter (PM2.5) collected over one year in Nanchang, Central China. Specific biomarkers and diagnostic ratios were applied to characterize OA sources and distribution patterns, while chemical mass balance (CMB) models and tracer-based approaches were used to estimate source contributions. Statistical analyses were conducted to investigate OA characteristics and drivers during winter pollution episodes. Notably, fatty acids, fatty alcohols, and saccharides predominated over lignin, resin products, sterols, glycerol, hydroxy acids, and aromatic acids, with molecular profiles indicating both anthropogenic and biogenic origins. Source apportionment results showed that primary organic carbon (POC) and primary OAs (POAs) contributed 58% of total organic carbon and 23% of PM2.5 mass, respectively, compared with 8% and 4% from secondary organic carbon (SOC) and secondary OAs (SOAs). Anthropogenic sources dominated, accounting for approximately 90% of POC and POAs as well as 60% of SOC and SOAs. Seasonal patterns revealed stronger biogenic influences in spring-summer, whereas anthropogenic emissions dominated in autumn-winter. Short-term winter episodes were characterized by rapid secondary formation, facilitated by elevated primary emissions and favorable oxidation conditions, including enhanced light intensity and nitrogen oxides." (lines 11–28, pages 1–2)

**Specific comments 11.** At lines 92-94, indicate the specific locations of the meteorological and gas sensors and their relation to the PM sampling site.

**Response:** Thank you for this helpful suggestion. We have added a precise description of the sensor locations and their spatial relationship to the PM sampling site. The specific content is as follows:

"In particular, the meteorological data were retrieved from the ground meteorological observation

station at Nanchang Changbei Airport, and gaseous pollutant data were obtained from the air quality monitoring station operated by the Jiangxi Academy of Forestry. These stations are the closest to the sampling site, located approximately 10 km and 2 km away, respectively. Both stations are ground-based, equipped with online monitoring instruments, and situated on open, flat terrain without surrounding buildings or obstructions. A detailed summary of the prevailing meteorological conditions and air quality during the sampling period is provided in Text S1 and Figure S2." (Lines 96–102, page 4).

**Specific comments 12.** Lines 99-100, include a reference to the IMPROVE protocol used for EC and OC analysis.

**Response:** Thank you — we have added a reference to the IMPROVE thermal/optical reflectance protocol and included a concise methods description.

Specifically: "The OC and EC concentrations in the PM2.5 samples were quantified using the Desert Research Institute Model 2001 Carbon Analyzer, following the thermal/optical reflectance protocol established by the Interagency Monitoring of Protected Visual Environments (IMPROVE) (Chow et al., 2007). A 1.0-cm2 filter sample was placed in a quartz boat in the analyzer and subjected to incremental heating at predetermined temperatures: 140 °C for OC1, 280 °C for OC2, 480 °C for OC3, and 580 °C for OC4 in a non-oxidizing helium atmosphere; and 580 °C for EC1, 740 °C for EC2, and 840 °C for EC3 in an oxidizing atmosphere containing 2% oxygen in helium." (Lines 104–110, pages 4–5).

**Specific comments 13.** Line 106, justification is needed for the use of n-alkanes as internal standards for polar compounds, especially because the alkanes do not undergo silylation derivitazation..

**Response:** Thank you for these valuable comments. We included the n-alkane standards mainly because the extracted species consist of polar and non-polar organic compounds, which are detected simultaneously by the instrument. Therefore, we added non-polar C13 n-alkanes as internal standards for quantitative analysis. The relevant explanation has been added in Section 2.2 "Chemical analyses" the to the manuscript. (**Lines 115–117, page 5**)

**Specific comments 14.** The statements about OC/EC values from 172-176 do not seem to consider secondary sources, or mixtures of sources. This seems contradictory with the other results in the study and should be removed.

**Response:** Thank you for these valuable comments. The relevant content has been removed from the manuscript.

**Specific comments 15.** Line 268, the authors should also consider reports of chemical degradation of levoglucosan in the atmosphere, for example, and how this may influence their source apportionment results.

Response: Thank you for this important comment. We have added a relevant discussion in Section

3.5 (Source apportionment of OC and aerosols) regarding the potential bias in source estimation caused by the wet and dry deposition as well as photochemical degradation of levoglucosan during transport. The specific details are as follows:

"Secondly, a fundamental assumption of the CMB model is that the selected markers representing various sources remain stable during atmospheric transport and do not undergo significant chemical transformation. However, truly conservative species are rare. For example, levoglucosan, commonly used as a biomass-burning marker and generally considered relatively stable (Wan et al., 2021), has nonetheless been shown to decrease during long-range transport through wet and dry deposition (Fu et al., 2011) and by photodegradation (Holmes and Petrucci, 2006; Stohl et al., 2007; Hoffmann et al., 2010). Analogous loss or alteration of other source tracers can bias CMB results, tending to underestimate contributions from sources whose markers are degraded or removed during transport." (lines 605–613, page 23)

**Specific comments 16.** In section 3.4.1, the authors report concentrations of C5 alkene triols. The authors should consider more up to date information available in the literature regarding this group of compounds (i.e. Frauenheim, et al. doi/10.1021/acs.estlett.2c00548). The majority of these "triols" have been demonstrated to be artifacts, and a structure with a ring rather than a double bond is major isomer.

**Response:** Thank you for bringing this important literature to our attention. We have incorporated the Frauenheim et al. (2022) findings that the C5-alkene triols with double bond structures are likely artifacts resulting from thermal decomposition during the GC–MS analysis. Therefore, in the revised manuscript, we have provided a detailed discussion on the impact of these artifacts on our source apportionment results. The specific discussion is as follows:

"Notably, the C5-alkene triols detected in our study predominantly exist as double-bonded "triol" including cis-2-methyl-1,3,4-trihydroxy-1-butene, compounds, 3-methyl-2,3,4-trihydroxy-1-butene, and trans-2-methyl-1,3,4-trihydroxy-1-butene. In fact, acid-catalyzed ring-opening reactions and isomerization of particle-phase IEPOX can also produce "diol" compounds, which typically exist as cyclic structures, trans-3-methyltetrahydrofuran-3,4-diol and cis-3-methyltetrahydrofuran-3,4-diol. These "diols" are also important components of C5-alkene triols (Li et al., 2013; Frauenheim et al., 2022). Recent work, however, indicates that C5-alkene triols detected via GC-MS are unlikely to originate solely from the acid-catalyzed ring-opening reactions and isomerization of IEPOX; instead, they may largely be artifacts produced by thermal decomposition during GC-MS analysis, with roughly 90% of the detected "triol" signal attributable to such artifacts (Frauenheim et al., 2022). If such artifacts indeed exist in the GC-MS measurements, it implies that the C5-alkene triol levels reported in our results are overestimated, potentially leading to an overall overestimation of isoprene-derived SOA tracers by approximately 30%. Despite the potential artifact issue in GC-MS detection, the contribution of C5-alkene triols to SOA should not be underestimated. These compounds are semi-volatile compounds that can volatilize back into the atmosphere from the particle phase, where they may undergo further oxidation by OH radicals. This oxidation process can significantly influence both the mass and composition of SOAs (Frauenheim et al., 2022 and 2024)." (lines 455–471, pages 17–18)

**Specific comments 17.** Figure 5, S6, F7. The text in the legends is very small and difficult to read. The important information that distinguishes the various sources is sub-scripted and difficult to see. Please enlarge the text in the legend to improve readability.

**Response:** Thank you for this helpful suggestion. We have revised Figures 5, S6 and F7 to improve readability by enlarging the legend text and increasing the size/clarity of subscripts that distinguish the source labels. High-resolution versions of the updated figures are included in the revised manuscript and Supplementary Information.

**Specific comments 18.** Lines 527-528, please explain how OC was converted into OA.

**Response:** Thank you for pointing this out. In Section 2.3 (Source apportionment methods), we have provided an explanation of how the conversion between OC and OA was handled when calculating the contributions of OC and OA using the CMB model and tracer-based methods.

Specifically: "Similarly, using the proportions of characteristic species in  $PM_{2.5}$  from these emission source profiles, the CMB model subsequently estimated each source's contribution to  $PM_{2.5}$ . In the primary emission profiles applied here, the PM-to-OC ratio ranged from 1.42 to 2.15." (lines 169–172, page 7)

"All marker selections, f-values, and the conversion coefficients between  $f_{SOA}$  and  $f_{SOC}$  derived from chamber studies are listed in Table S2, and the calculations assume that the chamber-derived SOC and SOA relationships are representative of the relevant atmospheric oxidation regimes, acknowledging associated uncertainties." (lines 190–193, pages 8)

The authors would like to express their sincere gratitude to the editor and reviewers for their valuable suggestions and comments, which have greatly improved the quality of this manuscript.

---

## Author Response (AR3)

**Response to the reviewer's recommendations point-by-point**

**Recommendation 1.** The added text regarding the alkene triols is very useful, except that the last two sentences from lines 468-471 appear to be tangential to the discussion. I suggest that these two sentences be removed to maintain the focus on the potential overestimation of isoprene-derived SOA.

**Response:** Thank you very much for your suggestion; we fully agree with it. In the revised manuscript, we have removed these two sentences.

**Recommendation 2.** The selected colors in figure 7 may not reproduce well. The pink text, in particular, is a bit light and difficult to read. Could this be made a darker color to improve readability? A darker shade of orange would also be helpful.

**Response:** Thank you for this suggestion. In the revised manuscript we updated Figure 7 with a darker orange (#663300) to improve contrast and applied the same change to Figure 6 and Figure S12 in the supplementary materials. The updated Figure 7 is as follows:

**Recommendation 3.** Figure 6 was difficult to understand. I read the caption a few times, but did not understand that the data shown is correlation coefficients of all things on the x-axis with OC.

Please clarify the caption to indicate that these are correlation coefficients between total measured OC and each individual primary and secondary source considered.

**Response:** Thank you for the suggestion. We have revised the Figure 6 caption to make clear that it reports correlation coefficients between measured OC and each individual primary and secondary source. The revised caption is below:

"Figure 6. Pearson correlations between measured OC and each individual POC and SOC source for the annual sampling period (a), winter pollution periods (b), winter remainder (c), and individual pollution episodes in winter (d–l)."

**Recommendation 4.** The Figure 5 legends are somewhat improved, but remain difficult to interpret. Because the first column of graphs are all primary sources, I suggest dropping "POC" from the legend and instead replacing these with the name of the source, i.e. biomass burning, coal combustion, etc. This will help the reader interpret the figure without having to decipher the caption and abbreviations. I suggest the same comment be applied to the second column of SOC plots, indicating that these corresponding to SOC precursors of napthalene, toluene, etc. For the third column that highlights the role of other sources, please label the legend as "Other OC" rather than OOC. I think OOC is confusing because of the commonality of OOA (oxygenated organic aerosol, which erroneously implies that this would be oxygenated organic carbon).

**Response:** Thank you for your valuable suggestion. We have revised Figure 5 accordingly; the updated figure is provided below. The same corrections have also been applied to Figures S7 and S9 in supplementary materials, which exhibited the same issues.

The authors would like to express their sincere gratitude to the editor and reviewers for their valuable suggestions and comments, which have greatly improved the quality of this manuscript.